# MiniGPT-v2: Large Language Model as a Unified Interface for Vision-Language Multi-task Learning

## Abstract

Large language models have shown their remarkable capabilities as a general interface for various language-related applications. Motivated by this, we target to build a unified interface for completing many vision-language tasks including image description, visual question answering, and visual grounding, among others. The challenge for achieving this is to use a single model for performing diverse vision-language tasks effectively with simple multi-modal instructions. To address this issue, we introduce MiniGPT-v2, a model can be treated a unified interface for better handling various vision-language tasks. We propose using unique identifiers for different tasks when training the model. These identifiers enable our model to distinguish each task instruction effortlessly and also improve the model learning efficiency for each task. After our three-stage training, the experimental results show that MiniGPT-v2 achieves strong performance on many visual question answering and visual grounding benchmarks compared to other vision-language generalist models. Our trained models and codes will be made available.

## 1 Introduction

Multi-modal Large Language Models (LLMs) have emerged as an exciting research topic with a rich set of applications in vision-language community, such as visual AI assistant, image captioning, visual question answering (VQA), and referring expression comprehension (REC). A key feature of multimodal large language models is that they can inherit advanced capabilities (e.g., logical reasoning, common sense, and strong language expression) from the LLMs (OpenAI, 2022; Touvron et al., 2023a;b; Chiang et al., 2023). When tuned with proper vision-language instructions, multi-modal LLMs, specifically vision-language models, demonstrate strong capabilities such as producing detailed image descriptions, generating code, localizing the visual objects in the image, and even perform multi-modal reasoning to better answer complicated visual questions (Zhu et al., 2023b; Liu et al., 2023b; Ye et al., 2023; Wang et al., 2023b; Chen et al., 2023b; Dai et al., 2023; Zhu et al., 2023a; Chen et al., 2023a; Zhuge et al., 2023). This evolution of LLMs enables interactions of visual and language inputs across communication with individuals and has been shown quite effective for building visual chatbots.

However, learning to perform multiple vision-language tasks effectively and formulating their corresponding multi-modal instructions present considerable challenges due to the complexities inherent among different tasks. For instance, given a user input *"tell me the location of a person"*, there are many ways to interpret and respond based on the specific task. In the context of the referring expression comprehension task, it can be answered with one bounding box location of the person. For the visual question answering, the model might describe the spatial location using human natural language. For person detection, the model might identify every spatial location of a human being. To alleviate this issue, we propose a task-oriented instruction training scheme to reduce the multi-modal instructional ambiguity, and a vision-language model, MiniGPT-v2. Specifically, we provide an unique task identifier token for each task. For example, we provide a *[vqa]* identifier token for training all the data samples from the visual question answering tasks. In total, we provide six different task identifiers during the model training stages.

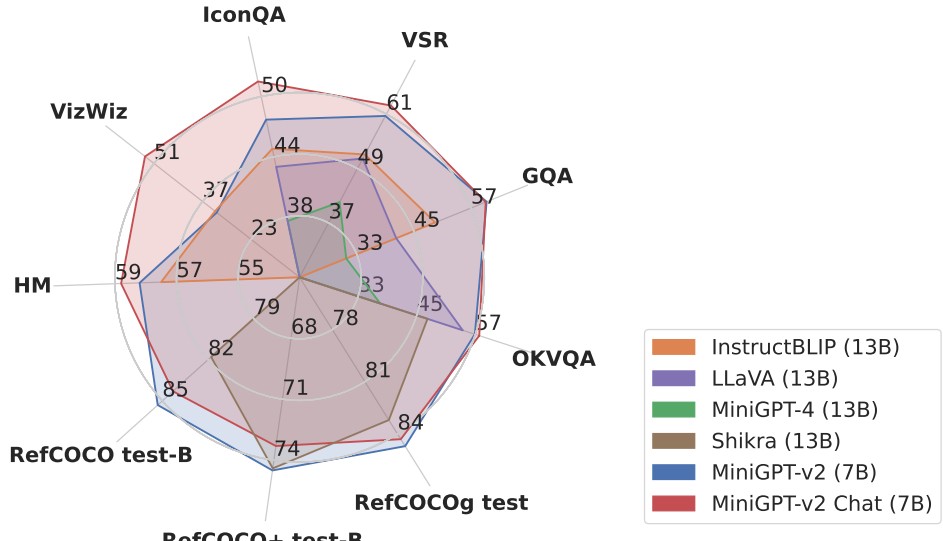

Figure 1: Our MiniGPT-v2 achieves state-of-the-art performances on a broad range of vision-language tasks compared with other generalist models.

Our model, MiniGPT-v2, has a simple architecture design. It directly takes the visual tokens from a ViT vision encoder (Fang et al., 2022) and project them into the feature space of a large language model (Touvron et al., 2023b). For better visual perception, we utilize high-resolution images (448x448) during training. But this will result in a larger number of visual tokens. To make the model training more efficient, we concatenate every four neighboring visual tokens into a single token, reducing the total number by 75%. Additionally, we utilize a three-stage training strategy to effectively train our model with a mixture of weakly-labeled, fine-grained image-text datasets, and multi-modal instructional datasets, with different training focus at each stage.

To evaluate the performance of our model, we conducted extensive experiments on diverse vision-language tasks, including (detailed) image/grounded captioning, vision question answering, and visual grounding. The results demonstrate that our MiniGPT-v2 can achieve SOTA or comparable performance on diverse benchmarks compared to previous vision-language generalist models, such as MiniGPT-4 (Zhu et al., 2023b), InstructBLIP (Dai et al., 2023), LLaVA (Liu et al., 2023b) and Shikra (Chen et al., 2023b). For example, our MiniGPT-v2 outperforms MiniGPT-4 by 21.7%, InstructBLIP by 11.2%, and LLaVA by 12.1% on the VSR benchmark (Liu et al., 2023a), and it also performs better than the previously established strong baseline, Shikra, in most validations on RefCOCO, RefCOCO+, and RefCOCOg. Our model establishes new state-of-the-art results on these benchmarks among vision-language generalist models, shown in Fig. 1.

## 2 RELATED WORK

We briefly review relevant works on Multi-task generalist models and multi-modal LLMs for visual aligning.

**Multi-task generalist models.** Recent years have witnessed significant advancements in vision-language learning, particularly in the development of multi-task generalist models (Hu & Singh, 2021; Yu et al., 2022; Lu et al., 2023; Singh et al., 2022; Zhang et al., 2021; Gan et al., 2020; Li et al., 2020; Yuan et al., 2021). These models act as versatile interfaces for a range of vision-language tasks. Unified I/O (Lu et al., 2022) integrates diverse tasks such as segmentation, depth estimation, and vision-language tasks. Florence (Yuan et al., 2021) expands the model training various representations, such as scene, object, images, videos, depths, and vision-language, via the web-scale image-text data training. BEIT-3 (Wang et al., 2022b) brings together various vision-language tasks through masked token prediction. FLAVA (Singh et al., 2022) pioneers a universal model by jointly pretraining on vision tasks, language tasks, and combined vision-language tasks.

ONE-PEACE (Wang et al., 2023a) aligns vision, language, and audio within a cohesive semantic framework. CLIP (Radford et al., 2021), Align Li et al. (2022), OpenCLIP (Ilharco et al., 2021), MetaCLIP (Xu et al., 2023a) align vision and language modalities in a shared semantic space, leveraging contrastive learning on extensive internet data.

**Multi-modal large language model (LLM).** Large language models (Radford et al., 2019; Devlin et al., 2018; Brown et al., 2020; OpenAI, 2022; Touvron et al., 2023a;b; Chowdhery et al., 2022; OpenAI, 2023) have achieved significant breakthroughs during the past few years. Their exceptional capabilities in generalization and representation have facilitated their expansion into the multi-modal domain by aligning visual inputs with LLMs. Initial efforts such as VisualGPT (Chen et al., 2022) and Frozen (Tsimpoukelli et al., 2021) used pre-trained language models to augment vision-language models for the image captioning and visual question answering. This initial exploration paved the way for subsequent vision-language research such as Flamingo (Alayrac et al., 2022) and BLIP-2 (Li et al., 2023b). More recently, GPT-4(V) (OpenAI, 2023) has been released and demonstrates many advanced multi-modal abilities, e.g., generating website code based on handwritten text instructions, based on its strong language model. Those demonstrated capabilities inspired other vision-language LLMs, including MiniGPT-4 (Zhu et al., 2023b), LLaVA (Liu et al., 2023b), mPLUG-Owl (Ye et al., 2023) and Otter Li et al. (2023a), which align the image inputs also with an advanced large language model with proper multi-modal instructional tuning. These vision-language models also showcase many advanced multi-modal capabilities similar to GPT-4(V). More recent developments include Vision-LLM (Wang et al., 2023b), Kosmos-2 (Peng et al., 2023), Shikra (Chen et al., 2023b), and our concurrent work, Qwen-VL (Bai et al., 2023). These models further explore visual grounding in the context of LLMs, pushing the boundaries of general multi-task vision-language modeling.

# 3 METHOD

In this section, we start by introducing our vision-language model, MiniGPT-v2, then discuss the basic idea of a multi-task instruction template with task identifier for training, and finally adapt our task identifier idea to achieve task-oriented instruction tuning.

## 3.1 MODEL ARCHITECTURE

Our proposed model architecture, MiniGPT-v2, is shown in Fig. 2. It consists of three components: a visual backbone, a linear projection layer, and a large language model. We describe each component as follows:

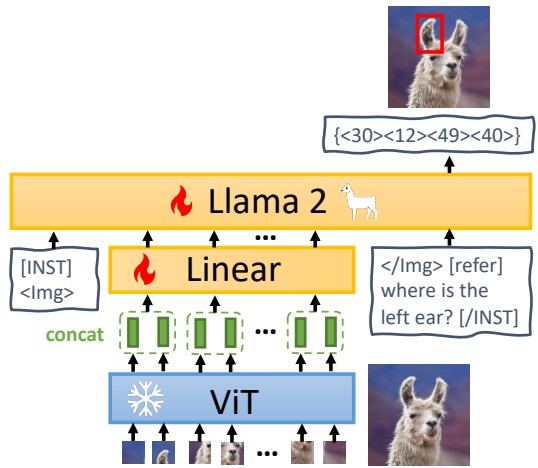

Figure 2: **Architecture of MiniGPT-v2.** The model takes a ViT visual backbone, which remains frozen during all training phases. We concatenate four adjacent visual output tokens from ViT backbone and project them into LLaMA-2 language model space via a linear projection layer.

**Visual backbone.** MiniGPT-v2 adapts the EVA (Fang et al., 2022) as our visual backbone model backbone. We freeze the visual backbone during the entire model training. We train our model with the image resolution 448x448, and we interpolate the positional encoding to scale with higher image resolution.

**Linear projection layer.** We aim to project all the visual tokens from the frozen vision backbone into the language model space. However, for higher-resolution images such as 448x448, projecting all the image tokens will result in a very long-sequence input (e.g., 1024 tokens) and significantly lowers the training and inference efficiency. To improve the efficiency, we simply concatenate 4 adjacent visual tokens in the embedding space and project them together into one single embedding in the same feature space of the large language model, thus reducing the number of visual input tokens by 4 times. With this operation, our MiniGPT-v2 can process high-resolution images much more efficiently during the training and inference stage.

**Large language model.** MiniGPT-v2 adopts the open-sourced LLaMA2-chat (7B) (Touvron et al., 2023b) as the language model backbone. In our work, the language model is treated as a unified interface for various vision-language inputs. We directly rely on the LLaMA-2 language tokens to perform various vision-language tasks. For the visual grounding tasks that necessitate the generation of spatial locations, we directly ask the language model to produce textual representations of bounding boxes to denote their spatial positions.

## 3.2 MULTI-TASK INSTRUCTION TEMPLATE

When training a single unified model for multiple different tasks such as visual question answering, image caption, referring expression, grounded image caption, and region identification, the multi-modal model might fail to distinguish each task by just aligning visual tokens to language models. For instance, when you ask "Tell me the spatial location of the person wearing a red jacket?", the model can either respond you the location in a bounding box format (e.g., $< X_{left} >< Y_{top} >< X_{right} >< Y_{bottom} >$) or describe the object location using natural language (e.g., upper right corner). To reduce such ambiguity and make each task easily distinguishable, we introduce task-specific tokens in our designed multi-task instruction template for training. We now describe our multi-task instruction template in more details.

**General input format.** We follow the LLaMA-2 conversation template design and adapt it for the multi-modal instructional template. The template is denoted as follows,

*[INST]  < ImageFeature> </Img> [Task Identifier] Instruction [/INST]*

In this template, *[INST]* is considered as the user role, and *[/INST]* is considered as the assistant role. We structure the user input into three parts. The first part is the image features, the second part is the task identifier token, and the third part is the instruction input.

**Task identifier tokens.** Our model takes a distinct identifier for each task to reduce the ambiguity across various tasks. As illustrated in Table 1, we have proposed six different task identifiers for visual question answering, image caption, grounded image captioning, referring expression comprehension, referring expression generation, and phrase parsing and grounding respectively. For vision-irrelevant instructions, our model does not use any task identifier token.

| Tasks | VQA | Caption | Grounded Caption | REC | REG | Object Parsing and Grounding |
|---|---|---|---|---|---|---|
| Identifiers | [vqa] | [caption] | [grounding] | [refer] | [identify] | [detection] |

Table 1: Task identifier tokens for 6 different tasks, including visual question answering, image captioning, , grounded image captioning, referring expression comprehension (REC), referring expression generation (REG), and object parsing and grounding (where the model extracts objects from the input text and determines their bounding box locations).

**Spatial location representation.** For tasks such as referring expression comprehension (REC), referring expression generation (REG), and grounded image captioning, our model is required to identify their spatial location of objects accurately. We represent the spatial location through the textual formatting of bounding boxes in our setting and do not use any new vocabulary tokens, specifically: "$\{< X_{left} >< Y_{top} >< X_{right} >< Y_{bottom} >\}$". Coordinates for X and Y are represented by integer values normalized in the range [0,100]. $< X_{left} >$ and $< Y_{top} >$ denote the x and y coordinate top-left corner of the generated bounding box, and $< X_{right} >$ and $< Y_{bottom} >$ denote the x and y coordinates of the bottom-right corner.

## 3.3 MULTI-TASK INSTRUCTION TRAINING

We now adapt our designed multi-task instruction template for instruction training. The basic idea is to take instruction with task-specific identifier token as input for task-oriented instruction training of MiniGPT-v2. When input instructions have task identifier tokens, our model will become more prone to multiple-task understanding during training. We train our model with task identifier instructions for better visual aligment in three stages. The first stage is to help MiniGPT-v2 build broad vision-language knowledge through many weakly-labeled image-text datasets, and high-quality fine-grained vision-language annotation datasets as well (where we will assign a high data

sampling ratio for weakly-labeled image-text datasets). The second stage is to improve the model with only fine-grained data for multiple tasks. The third stage is to finetune our model with more multi-modal instruction and language datasets for answering diverse multi-modal instructions better and behaving as a multi-modal chatbot. The datasets used for training at each stage are listed in the Table 2.

| Data types | Dataset | Stage 1 | Stage 2 | Stage 3 |
|---|---|---|---|---|
| Weakly-labeled | GRIT-20M (REC and REG), LAION, CC3M, SBU | ✓ | ✗ | ✗ |
| Grounded caption | GRIT-20M | ✓ | ✗ | ✗ |
| Caption | COCO caption, TextCaps | ✓ | ✓ | ✓ |
| REC | RefCOCO, RefCOCO+, RefCOCOg, Visual Genome | ✓ | ✓ | ✓ |
| REG | RefCOCO, RefCOCO+, RefCOCOg | ✓ | ✓ | ✓ |
| VQA | GQA, VQAv2, OCR-VQA, OK-VQA, AOK-VQA | ✓ | ✓ | ✓ |
| Multimodal instruction | LLaVA dataset, Flickr30k, Multi-task conversation | ✗ | ✗ | ✓ |
| Langauge dataset | Unnatural Instructions | ✗ | ✗ | ✓ |

Table 2: The training datasets used for our model three-stage training.

**Stage 1: Pretraining.** To have broad vision-language knowledge, our model is trained on a mix of weakly-labeled and fine-grained datasets. We give a high sampling ratio for weakly-labeled datasets to gain more diverse knowledge in the first-stage.

For the weakly-labeled datasets, we use LAION (Schuhmann et al., 2021), CC3M (Sharma et al., 2018), SBU (Ordonez et al., 2011), and GRIT-20M from Kosmos v2 (Peng et al., 2023) that built the dataset for referring expression comprehension (REC), referring expression generation (REG), and grounded image captioning. The format for grounded image caption is represented like this: *a <p>wooden table</p>{$<X_{left}><Y_{top}><X_{right}><Y_{bottom}>$} in the center of the room.*

For fine-grained datasets, we use datasets like COCO caption (Lin et al., 2014) and TextCaps (Sidorov et al., 2020) for image captioning, RefCOCO (Kazemzadeh et al., 2014), RefCOCO+ (Yu et al., 2016), and RefCOCOg (Mao et al., 2016) for REC. For REG, we restructured the data from ReferCOCO and its variants, reversing the order from phrase → bounding boxes to bounding boxes → phrase. For VQA datasets, our training takes a variety of datasets, such as GQA (Hudson & Manning, 2019), VQA-v2 (Goyal et al., 2017), OCR-VQA (Mishra et al., 2019), OK-VQA (Marino et al., 2019), and AOK-VQA (Schwenk et al., 2022).

**Stage 2: Multi-task training.** To improve the performance of MiniGPT-v2 on each task, we only focus on using fine-grained datasets to train our model at this stage. We exclude the weakly-supervised datasets such as GRIT-20M and LAION from stage-1 and update the data sampling ratio according to frequency of each task. This strategy enables our model to prioritize high-quality aligned image-text data for superior performance across various tasks.

**Stage 3: Multi-modal instruction tuning.** Subsequently, we focus on tuning our model with more multi-modal instruction dataset and enhance its conversation ability as a chatbot. We continue using the datasets from the second stage, and add instructional datasets, including LLaVA (Liu et al., 2023b), Flickr30k dataset (Plummer et al., 2015), our constructed mixing multi-task dataset, and the language dataset, Unnatural Instruction (Honovich et al., 2022). We give a lower data sampling ratio for the fine-grained datasets from stage-2 and a higher data sampling ratio for the new instruction datasets.

**– LLaVA instruction data.** We add the multi-modal instruction tuning datasets, including the detailed descriptions and complex reasoning from LLaVA (Liu et al., 2023b), with 23k and 58k data examples respectively.

**– Flicker 30k.** After the second-stage training, our MiniGPT-v2 can effectively generate the grounded image caption. Nevertheless, these descriptions tend to be short and often cover very few number of visual objects. This is because the GRIT-20M dataset from KOSMOS-v2 (Peng et al., 2023) that our model was trained with, features a limited number of grounded visual objects in each caption, and our model lacks proper multi-modal instruction tuning to teach it to recognize more visual objects. To improve this, we fine-tune our model using the Flickr30k dataset (Plummer et al., 2015), which provides more contextual grounding of entities within its captions.

We prepare the Flickr30k dataset in two distinct formats for training our model to perform grounded image caption and a new task "object parsing and grounding":

1) **Grounded image caption.** We select captions with a minimum of five grounded phrases, containing around 3k samples, and we directly instruct the model to produce the grounded image caption. *a <p>wooden table</p>{<$X_{left}$><$Y_{top}$><$X_{right}$><$Y_{bottom}$>} in the center of the room.* The format for grounded image caption is

2) **Object parsing and grounding.** This new task is to parse all the objects from an input caption and then ground each object. To enable this, we simply use the task identifier*[detection]* to differentiate this capability from other tasks. Also, we use Flickr30k to construct two types of instruction datasets: caption→ grounded phrases and phrase → grounded phrase, each containing around 3k and 4k samples. Then we prompt our model with the instruction: *"[detection] description"*, the model will directly parse the objects from the input image description and also ground the objects into bounding boxes.

– **Mixing multi-task dataset.** After extensive training with single-round instruction-answer pairs, the model might not handle multiple tasks well during multi-round conversations since the context becomes more complex. To alleviate this situation, we create a new multi-round conversation dataset by mixing the data from different tasks. We include this dataset into our third-stage model training.

– **Unnatural instruction.** The conversation abilities of language model can be reduced after extensive vision-language training. To fix this, we add the language dataset, Unnatural Instruction (Honovich et al., 2022) into our model's third-stage training for helping recover the language generation ability.

## 4 EXPERIMENTS

In this section, we present experimental settings and results. We primarily conduct experiments on (detailed) image/grounded captioning, vision question answering, and visual grounding tasks, including referring expression comprehension. We present both quantitative and qualitative results.

**Implementation details.** Throughout the entire training process, the visual backbone of MiniGPT-v2 remains frozen. We focus on training the linear projection layer and efficient finetuning the language model using LoRA (Hu et al., 2021). With LoRA, we finetune $\mathcal{W}_q$ and $\mathcal{W}_v$ via low-rank adaptation. In our implementation, we set the rank, $r = 64$. We trained the model with an image resolution of 448x448 during all stages. During each stage, we use our designed multi-modal instructional templates for various vision-language tasks during the model training.

**Training and hyperparameters.** We use AdamW optimizer with a cosine learning rate scheduler to train our model. In the initial stage, we train on 8xA100 GPUs for 400,000 steps with a global batch size of 96 and an maximum learning rate of 1e-4. This stage takes around 90 hours. During the second stage, the model is trained for 50,000 steps on 4xA100 GPUs with a maximum learning rate of 1e-5, adopting a global batch size of 64, and this training stage lasts roughly 20 hours. For the last stage, training is executed for another 50,000 steps on 4xA100 GPUs, using a global batch size of 24 and this training stage took around 10 hours, maintaining the same maximum learning rate of 1e-5.

### 4.1 QUANTITATIVE EVALUATION

**Dataset and evaluation metrics.** We evaluate our model across a range of VQA and visual grounding benchmarks. For VQA benchmarks, we consider OKVQA (Schwenk et al., 2022), GQA (Hudson & Manning, 2019), VSR (Liu et al., 2023a), IconVQA (Lu et al., 2021), VizWiz (Gurari et al., 2018), HatefulMemes (Kiela et al., 2020), and TextVQA (Singh et al., 2019). For visual grounding, we evaluate our model on RefCOCO (Kazemzadeh et al., 2014) and RefCOCO+(Yu et al., 2016), and RefCOCOg(Mao et al., 2016) benchmarks. More details about the dataset and evaluation metrics can be found in the appendix.

**Visual question answering results.** Table 3 presents our experimental results on multiple VQA benchmarks. Our results compare favorably to baselines including MiniGPT-4 (Zhu et al., 2023b), Shikra (Chen et al., 2023b), LLaVA (Liu et al., 2023b), mPLUG-Owl (Ye et al., 2023), Otter (Li

| Method | Grounding | OKVQA | GQA | VSR (zero-shot) | TextVQA (zero-shot) | IconVQA (zero-shot) | VizWiz (zero-shot) | HM (zero-shot) |
|---|---|---|---|---|---|---|---|---|
| Flamingo-9B | ✗ | 44.7 | - | 31.8 | - | - | 28.8 | 57.0 |
| BLIP-2 (13B) | ✗ | 45.9 | 41.0 | 50.9 | 42.5 | 40.6 | 19.6 | 53.7 |
| mPLUG-Owl (7B) | ✗ | 22.9 | 14.0 | 11.6 | 38.8 | 11.6 | 39.0 | - |
| Otter (7B) | ✗ | 49.0 | 38.1 | 6.4 | 21.5 | 38.2 | 50.0 | - |
| InstructBLIP (13B) | ✗ | - | 49.5 | 52.1 | 50.7 | 44.8 | 33.4 | 57.5 |
| MiniGPT-4 (13B) | ✗ | 37.5 | 30.8 | 41.6 | 19.4 | 37.6 | - | - |
| LLaVA (13B) | ✗ | 54.4 | 41.3 | 51.2 | 38.9 | 43.0 | - | - |
| Shikra (13B) | ✓ | 47.2 | - | - | - | - | - | - |
| Ours (7B) | ✓ | **56.9** | **60.3** | 60.6 | 51.9 | 47.7 | 30.3 | 58.2 |
| Ours (7B)-chat | ✓ | 55.9 | 58.8 | **63.3** | **52.3** | **49.4** | **53.0** | **59.5** |

Table 3: **Results on multiple VQA tasks.** We report top-1 accuracy for each task. Grounding column indicates whether the model incorporates visual localization capability. The best performance for each benchmark is indicated in **bold**.

| Method | Model types | RefCOCO | | | RefCOCO+ | | | RefCOCOg | | Avg |
|---|---|---|---|---|---|---|---|---|---|---|
| | | val | test-A | test-B | val | test-A | test-B | val | test | |
| UNINEXT | Specialist models | 92.64 | 94.33 | 91.46 | 85.24 | 89.63 | 79.79 | 88.73 | 89.37 | 88.90 |
| G-DINO-L | | 90.56 | 93.19 | 88.24 | 82.75 | 88.95 | 75.92 | 86.13 | 87.02 | 86.60 |
| VisionLLM-H | Generalist models | - | 86.70 | - | - | - | - | - | - | - |
| OFA-L | | 79.96 | 83.67 | 76.39 | 68.29 | 76.00 | 61.75 | 67.57 | 67.58 | 72.65 |
| Shikra (7B) | | 87.01 | 90.61 | 80.24 | 81.60 | 87.36 | 72.12 | 82.27 | 82.19 | 82.93 |
| Shikra (13B) | | 87.83 | 91.11 | 81.81 | **82.89** | **87.79** | 74.41 | 82.64 | 83.16 | 83.96 |
| Ours (7B) | | **88.69** | **91.65** | **85.33** | 79.97 | 85.12 | **74.45** | 84.44 | 84.66 | **84.29** |
| Ours (7B)-chat | | 87.18 | 90.51 | 84.57 | 78.69 | 84.25 | 73.23 | 81.88 | 83.25 | 82.95 |

Table 4: **Results on referring expression comprehension tasks.** Our MiniGPT-v2 outperforms many VL-generalist models including VisionLLM (Wang et al., 2023b), OFA (Wang et al., 2022a) and Shikra (Chen et al., 2023b) and reduces the accuracy gap comparing to specialist models including UNINEXT (Yan et al., 2023) and G-DINO (Liu et al., 2023c).

et al., 2023a) and InstructBLIP (Dai et al., 2023) across all the VQA tasks. The results for mPLUG-Owl, Otter and MiniGPT-4 are borrowed from (Xu et al., 2023b) For example, on QKVQA, our MiniGPT-v2 outperforms MiniGPT-4, Shikra, LLaVA, mPLUG-Owl, Otter and BLIP-2 by 19.4%, 9.7%, 2.5%, 34%, 7.9% and 11%. These results indicate the strong visual question answering capabilities of our model. Furthermore, we find that our MiniGPT-v2 (chat) variant shows higher performance than the version trained after the second stage. On VSR, TextVQA, IconVQA, VizWiz, and HM, MiniGPT-v2 (chat) outperforms MiniGPT-v2 by 2.7%, 0.4%, 1.7%, 12.1%, and 1.3%. We believe that the better performance can be attributed to the improved language skills during the third-stage training, which is able to benefit visual question comprehension and response, especially on VizWiz with 12.1% top-1 accuracy increase.

**Referring expression comprehension results.** Table 4 compares our model to baselines on REC benchmarks. Our MiniGPT-v2 shows strong REC performance on RefCOCO, RefCOCO+, and RefCOCOg, performing better than other vision-language generalist models. MiniGPT-v2 outperforms OFA-L (Wang et al., 2022a) by over 8% accuracy across all tasks of RefCOCO/RefCOCO+/RefCOCOg. Compared with a strong baseline, Shikra (13B) (Chen et al., 2023b), our model still shows better results, e.g., 84.29% vs 83.96% accuracy in average. These results provide direct evidence for the competing visual grounding capabilities of MiniGPT-v2. Although our model underperforms specialist models, the promising performance indicates its growing competence in visual grounding.

**Ablation on task identifier.** We conduct ablation studies on the effect of the task identifier on the performance of MiniGPT-v2. We compare our model with the variant without using task identifiers on VQA benchmarks. Both models were trained on 4xA100 GPUs for 24 hours with an equal number of training steps for multiple vision-language tasks. Results in Table 5 demonstrate the performance on multiple VQA benchmarks and consistently show that token identifier training benefits the overall performance of MiniGPT-v2. Specifically, our MiniGPT-v2 with task-oriented instruction training achieves 1.2% top-1 accuracy improvement on average. These ablation results

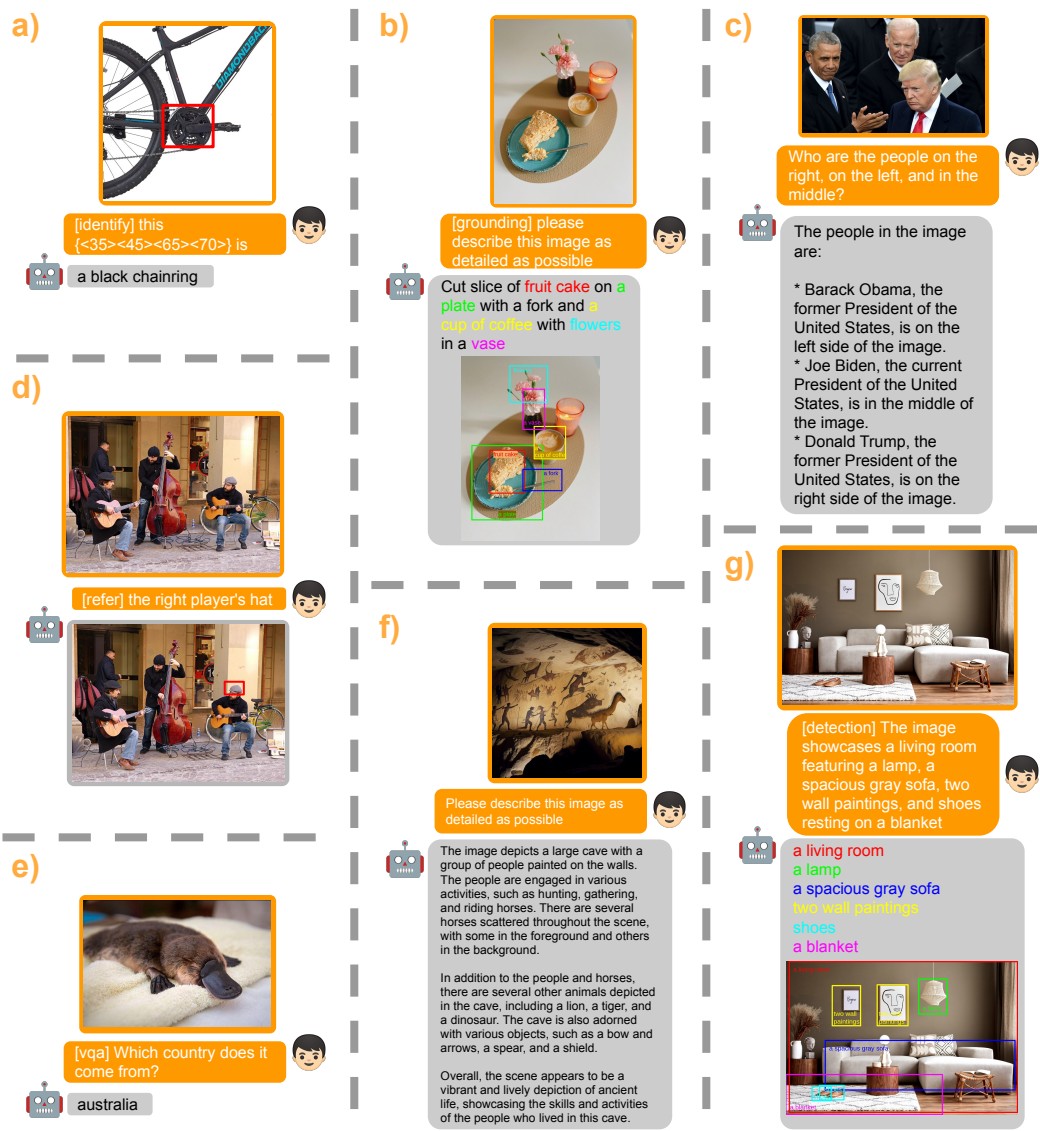

Figure 3: **Examples for various multi-modal capabilities of MiniGPT-v2.** We showcase that our model is capable of completing multiple tasks such as referring expression comprehension, referring expression generation, detailed grounded image caption, visual question answering, detailed image description, and directly parsing phrase and grounding from a given input text.

can validate the clear advantage of adding task identifier tokens and support the use of multi-task identifiers for multi-task learning efficiency.

| | OKVQA | GQA | VizWiz | VSR | IconVQA | HM | Average |
|---|---|---|---|---|---|---|---|
| Ours w/o task identifier | 50.5 | 53.4 | 28.6 | 57.5 | 44.8 | 56.8 | 48.6 |
| Ours | **52.1** | **54.6** | **29.4** | **59.9** | **45.6** | **57.4** | **49.8** |

Table 5: Task identifier ablation study on VQA benchmarks. With task identifier during the model training can overall improve VQA performances from multiple VQA benchmarks

**Hallucination.** We measure the hallucination of our model on image description generation and compare the results with other vision-language baselines, including MiniGPT-4 (Zhu et al., 2023b), mPLUG-Owl (Ye et al., 2023), LLaVA (Liu et al., 2023b), and MultiModal-GPT (Gong et al., 2023). Following the methodology from (Li et al., 2023c), we use CHAIR (Rohrbach et al., 2018) to assess hallucination at both object and sentence levels. As shown in Table 6, we find that our MiniGPT-v2 tends to generate the image description with reduced hallucination compared to other baselines. We have evaluated three types of prompts in MiniGPT-v2. First, we use the prompt *generate a brief description of the given image* without any specific task identifier

| Method | CHAIR$_I \downarrow$ | CHAIR$_S \downarrow$ | Len |
|---|---|---|---|
| MiniGPT-4 | 9.2 | 31.5 | 116.2 |
| mPLUG-Owl | 30.2 | 76.8 | 98.5 |
| LLaVA | 18.8 | 62.7 | 90.7 |
| MultiModal-GPT | 18.2 | 36.2 | 45.7 |
| MiniGPT-v2 (long) | 8.7 | 25.3 | 56.5 |
| MiniGPT-v2 (grounded) | 7.6 | 12.5 | 18.9 |
| MiniGPT-v2 (short) | **4.4** | **7.1** | **10.3** |

Table 6: **Results on hallucination.** We evaluate the hallucination of MiniGPT-v2 with different instructional templates and output three versions of captions for evaluation. For the "long" version, we use the prompt *generate a brief description of the given image*. For the "grounded" version, the instruction is *[grounding] describe this image in as detailed as possible*. For the "short" version, the prompt is *[caption] briefly describe the image*.

which tends to produce more detailed image descriptions. Then we provide the instruction prompt *[grounding] describe this image in as detailed as possible* for evaluating grounded image captions. Lastly, we prompt our model with *[caption] briefly describe the image*. With these task identifiers, MiniGPT-v2 is able to produce a variety of image descriptions with different levels of hallucination. As a result, all these three instruction variants have lower hallucination than our baseline, especially with the task specifiers of *[caption]* and *[grounding]*.

## 4.2 QUALITATIVE RESULTS

We now provide the qualitative results for a complementary understanding of our model's multi-modal capabilities. Some examples can be seen in Fig. 3. Specifically, we demonstrated various abilities in the examples including a) object identification; b) detailed grounded image captioning; c) visual question answering; d) referring expression comprehension; e) visual question answering under task identifier; f) detailed image description; g) object parsing and grounding from an input text. More qualitative results can be found in the Appendix. These results demonstrate that our model has competing vision-language understanding capabilities. Moreover, notice that we train our model only with a few thousand of instruction samples on object parsing and grounding tasks at the third-stage, and our model can effectively follow the instructions and generalize on the new task. This indicates that our model has the flexibility to adapt on many new tasks.

## 5 CONCLUSION AND DISCUSSION

In this paper, we introduce MiniGPT-v2, a multi-modal LLM that can serve as a unified interface for various vision-language multi-tasking learning. To develop a single model capable of handling multiple vision-language tasks, we propose using distinct identifiers for each task during the training and inference. These identifiers help our model easily differentiate various tasks and also improve the learning efficiency. Our MiniGPT-v2 achieves strong results across many visual question answering and referring expression comprehension benchmarks. We also found that our model can efficiently adapt to new vision-language task, which suggests that MiniGPT-v2 has many potential applications in vision-language community.

However, our model still occasionally shows hallucinations when generating the image description, visual grounding or answering visual questions. e.g., our model may sometimes produce descriptions of non-existent visual objects or generate inaccurate visual locations of grounded objects. We believe aligning the models with more high-quality image-text aligned data, and integrating with a stronger vision backbone and large language model hold the potential for alleviating this issue.

In the appendix, we provide more qualitative results that are generated from our model to demonstrate the vision-language multi-tasking capabilities.

## A  EVALUATION METRICS

**Visual Question Answering (VQA):** For the VQA benchmarks, we adopted the standard open-ended VQA evaluation metrics. This involves comparing the model's response directly with the ground truth, and we report the top-1 accuracy as a measure of performance.

**RefCOCO/+/g:** In assessing visual grounding, we implemented a two-step process. Initially, our model generates a bounding box for each answer, which is then evaluated through the Intersection over Union (IoU) method. We specifically measure the overlap between the generated and ground-truth bounding boxes, considering overlaps greater than 0.5 as indicative of correct grounding.

**Hallucination Evaluation:** To address the critical aspect of hallucination in generated responses, we employed $CHAIR_i$ and $CHAIR_s$ metrics. $CHAIR_i$ calculates the proportion of hallucinated objects relative to all mentioned objects, offering insight into the frequency of hallucination occurrences. Conversely, $CHAIR_s$ assesses the ratio of captions containing hallucinated objects to the total number of captions, providing a broader view of the model's overall tendency towards hallucination. Here is how $CHAIR_i$ and $CHAIR_s$ are calculated

$$CHAIR_i = \frac{|\{\text{hallucinated objects}\}|}{|\{\text{all mentioned objects}\}|},$$

$$CHAIR_s = \frac{|\{\text{captions with hallucinated objects}\}|}{|\{\text{all captions}\}|},$$

## B  DATASET DETAILS

**Training data.** Here we demonstrate the statistics

- GQA (Hudson & Manning, 2019): VQA on scene understanding and reasoning. 22M questions.
- VQAv2 (Goyal et al., 2017): VQA on natural images. 443,757 questions.
- OCR-VQA (Mishra et al., 2019): VQA on images of book covers. 1,002,146 questions.
- OK-VQA (Schwenk et al., 2022): VQA on natural images requiring outside knowledge. 14K questions.
- AOK-VQA (Schwenk et al., 2022): Augmented VQA on natural images requiring outside knowledge. 25K questions.
- LLaVA (Liu et al., 2023b) instruction: 23k detailed descriptions and 58k complex reasoning examples.
- LAION (Schuhmann et al., 2021): CLIP-filtered 400 million image-text pairs.
- CC3M (Sharma et al., 2018): It contains 3.3M web images annotated with captions.
- SBU (Ordonez et al., 2011): A large captioned photo collection with 1 million images on Flickr.
- COCO caption (Lin et al., 2014): It consists of a half million captions describing over 330,000 images.
- TextCaps (Sidorov et al., 2020): It contains 28,408 images from OpenImages, 142,040 captions.
- Flickr30K (Plummer et al., 2015) contains 31,783 images.
- RefCOCO (Kazemzadeh et al., 2014) contains 142,209 referring expression for 50,000 objects in 19,994 images.

- RefCOCOg (Mao et al., 2016) contains 85,474 referring expression for 54,822 objects in 26,711 images
- RefCOCO+ (Yu et al., 2016) contains 141,564 referring expression for 49,856 objects in 19,992 images
- GRIT 20M (Peng et al., 2023) it contains around 20M grounded image caption.

## C    INSTRUCTION TEMPLATE FOR VARIOUS VISION-LANGUAGE TASKS

**RefCOCO/RefCOCO+/RefCOCOg:** *[refer] give me the location of* {*question*}

**VizWiz:** *[vqa] Based on the image, respond to this question with a short answer:* {*question*} *and reply 'unanswerable' if you could not answer it*

**Hateful Meme:** *[vqa] This is an image with:* {*question*} *written on it. Is it hateful? Answer:*

**VSR:** *[vqa] Based on the image, is this statement true or false?* {*question*}

**IconQA, GQA, OKVQA:** *[vqa] Based on the image, respond to this question with a short answer:* {*question*}

## D    MINIGPT-V2 CONVERSATION EVALUATION

We evaluate MiniGPT-v2 under a multi-round conversation, and we demonstrate the conversation in the Fig. 4

## E    ADDITIONAL QUALITATIVE RESULTS

To study how well our model is able to take visual input and answer questions based on task-oriented identifier, we use our model to perform multiple vision-language tasks including grounded image captioning in Fig. 5, Fig. 6, Fig. 7 and Fig. 8; Object parsing and grounding in Fig. 9, Fig. 10, Fig. 11 and Fig. 12; Referring expression comprehension in Fig. 13, Fig. 14, Fig. 15 and Fig. 16; Object identification in Fig. 17, Fig. 18, Fig. 19 and Fig. 20.

For each task, we share 4 examples for showing the vision-language capabilities of our model. The results in the demo provide direct evidence for the competing visual understanding capabilities of MiniGPT-v2 on multiple vision-language tasks. For example, in the cases of grounded caption, our model is able to give correct grounded image caption with detailed spatial locations of objects. In the cases of identify, the model also generates our expected object names. MiniGPT-v2 can understand the new scenes and follow the question identifier to respond. But we also need to note that our model still has some hallucination e.g., In Fig. 7, several persons are not grounded accurately, and in Fig. 8, there does not exist a vase in the image.

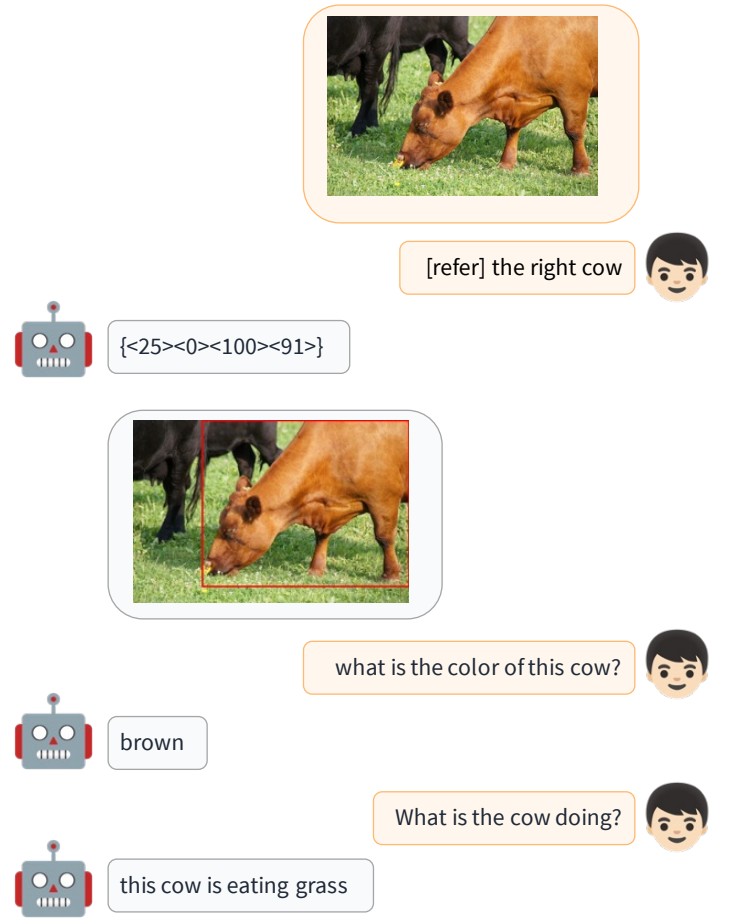

Figure 4: Conversation evaluation.

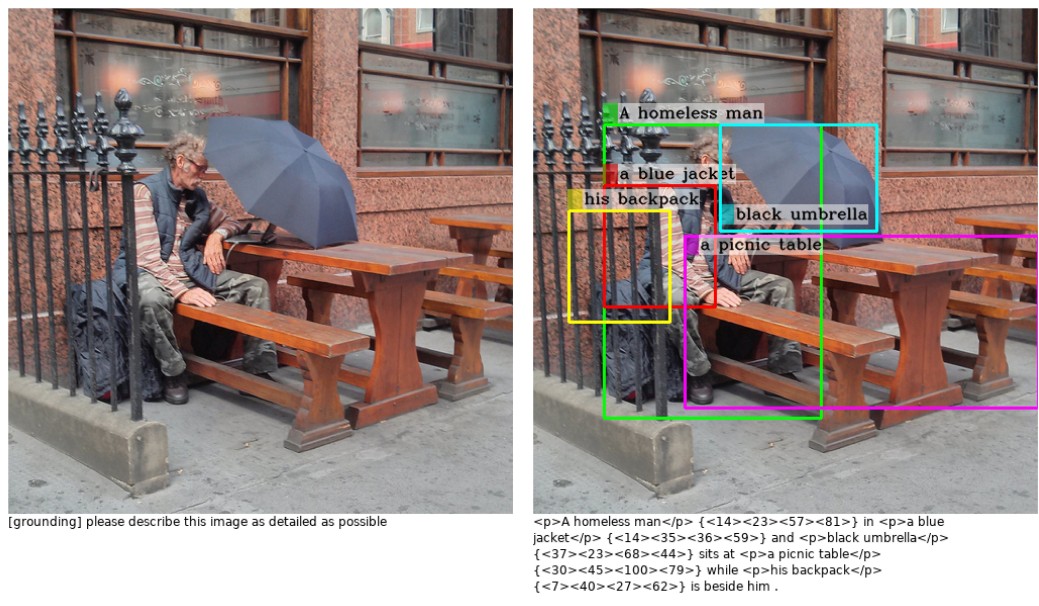

Figure 5: Detail grounded image caption example.

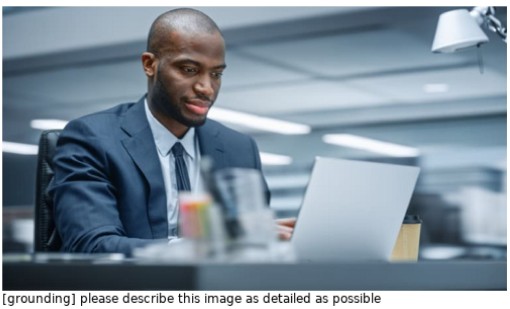 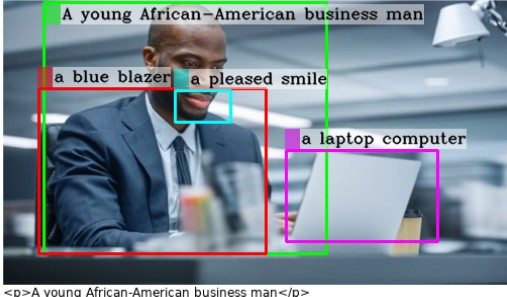

Figure 6: Detail grounded image caption example

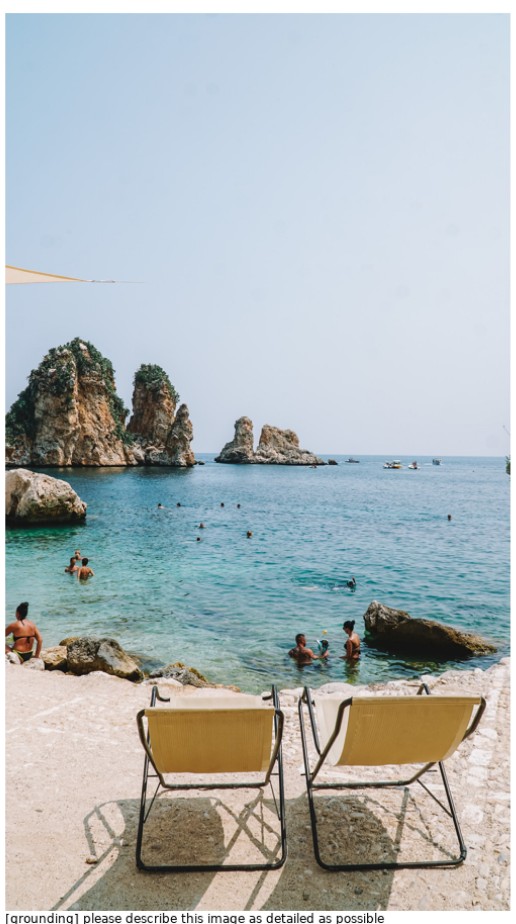 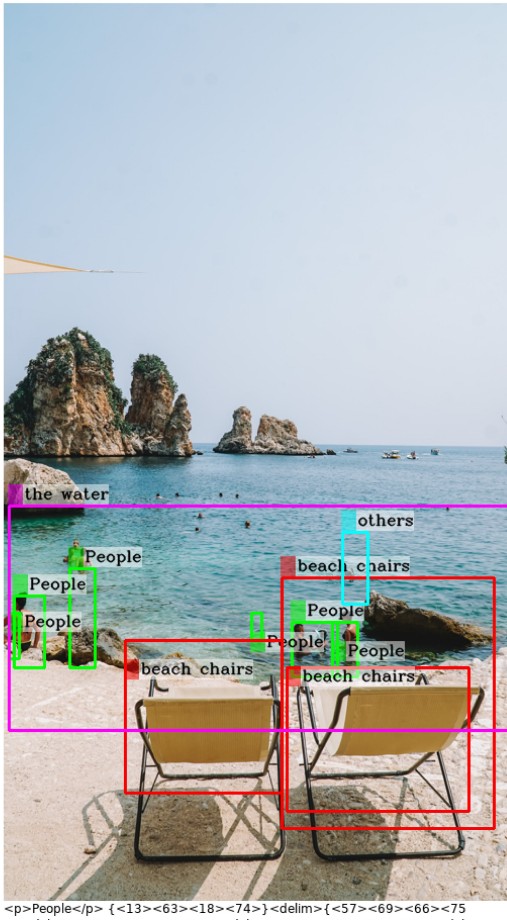

Figure 7: Detail grounded image caption example

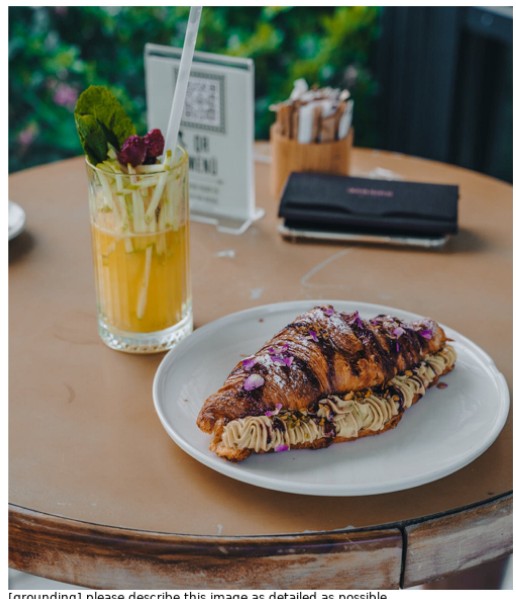 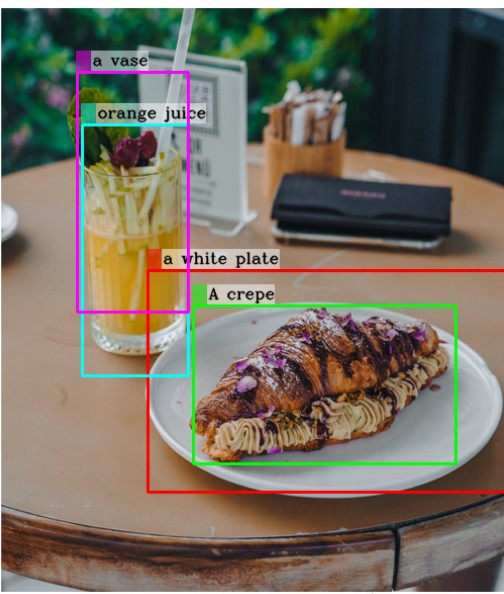

[grounding] please describe this image as detailed as possible

<p>A crepe</p> {<38><51><90><78>} sits on <p>a white plate</p> {<29><45><100><83>} near a cup of <p>orange juice</p> {<16><20><37><63>} and <p>a vase</p> {<15><11><37><52>} with a stem in the background .

Figure 8: Detail grounded image caption example

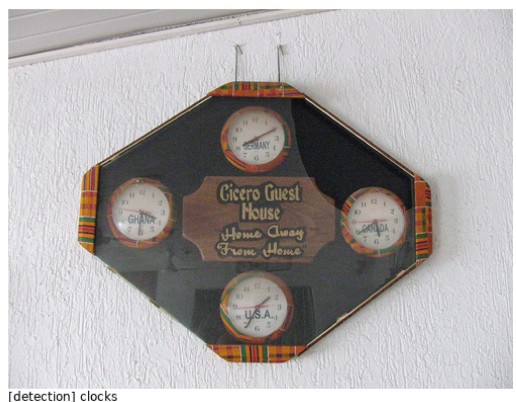 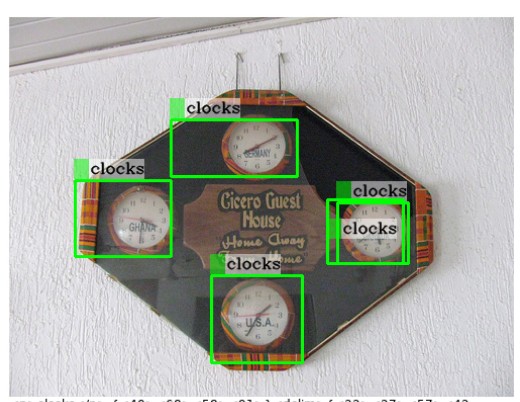

[detection] clocks

<p>clocks</p> {<40><68><58><91>}<delim>{<32><27><57><42>}<delim>{<13><43><32><63>}<delim>{<65><49><79><65>}<delim>{<63><48><78><65>}

Figure 9: Object parsing and grounding example

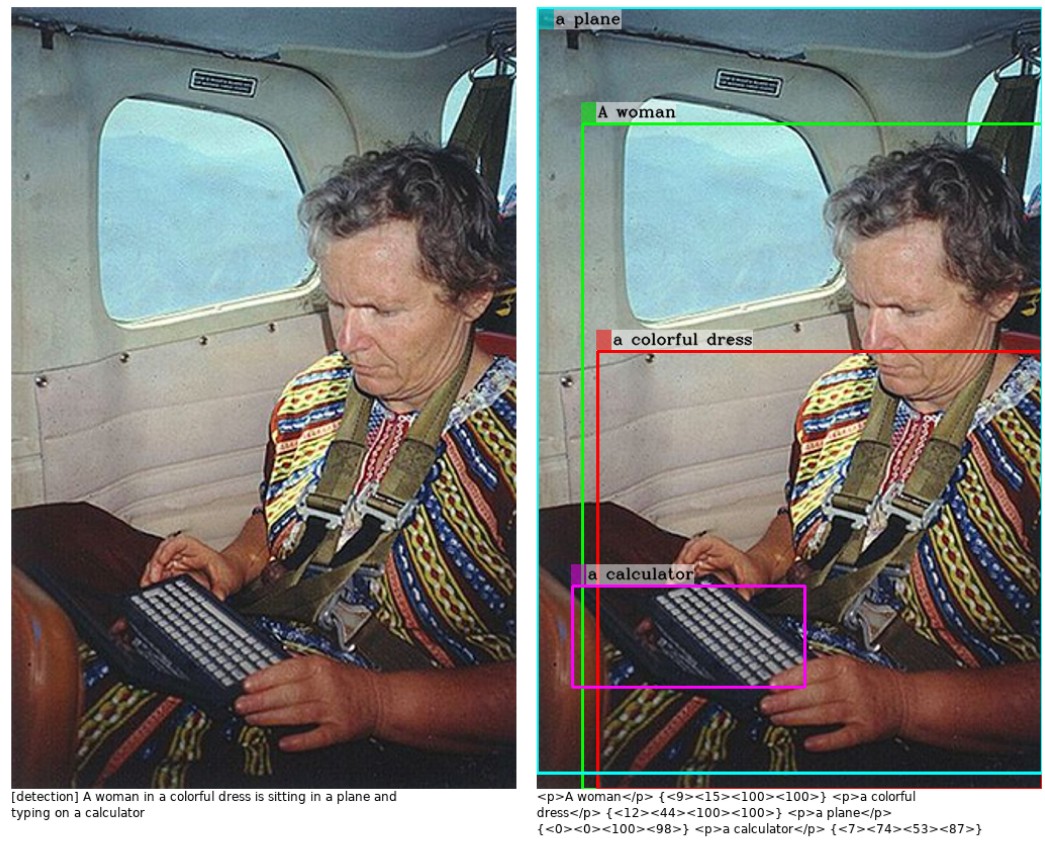

[detection] A woman in a colorful dress is sitting in a plane and typing on a calculator

<p>A woman</p> {<9><15><100><100>} <p>a colorful dress</p> {<12><44><100><100>} <p>a plane</p> {<0><0><100><98>} <p>a calculator</p> {<7><74><53><87>}

Figure 10: Object parsing and grounding example

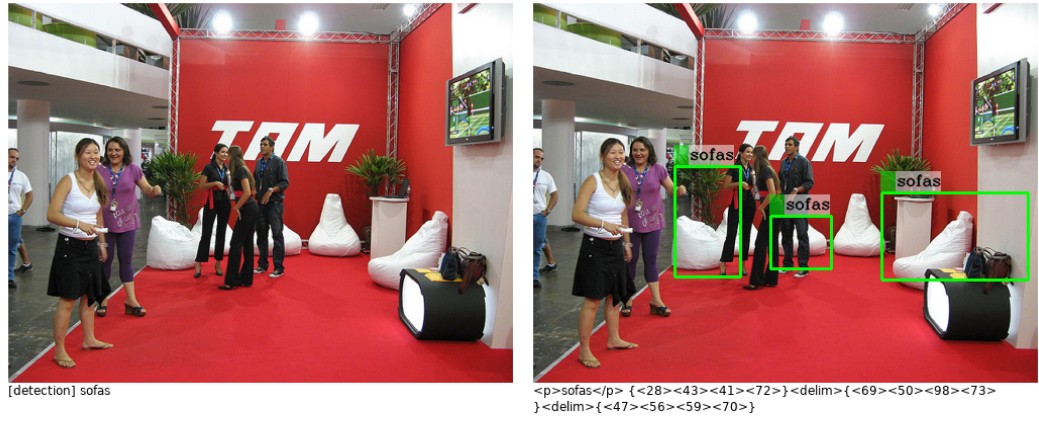

[detection] sofas

<p>sofas</p> {<28><43><41><72>}<delim>{<69><50><98><73>}<delim>{<47><56><59><70>}

Figure 11: Object parsing and grounding example

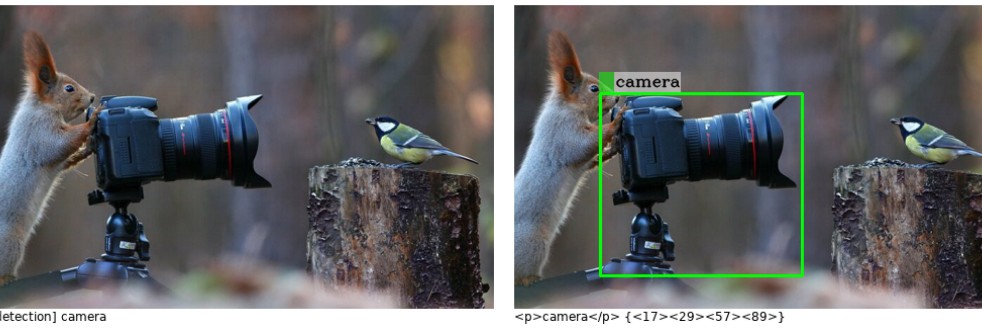

Figure 12: Object parsing and grounding example

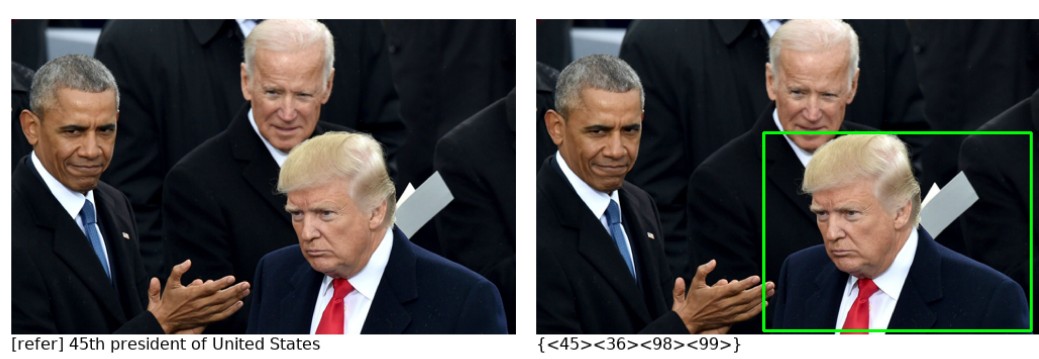

Figure 13: Referring expression comprehension example

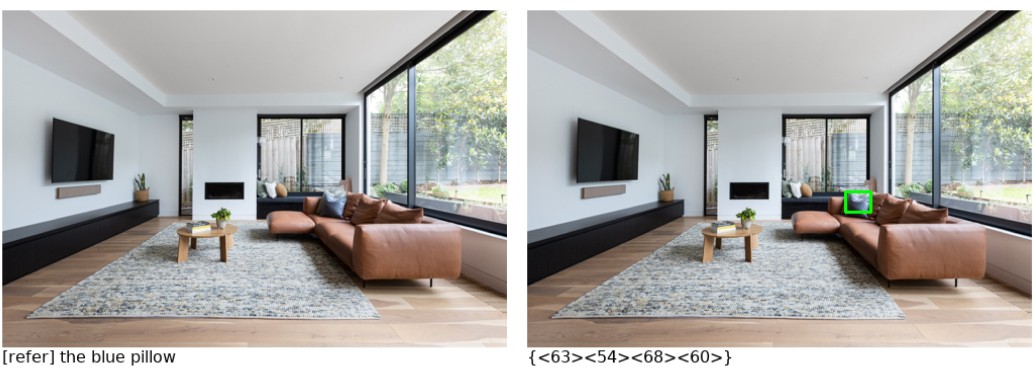

Figure 14: Referring expression comprehension example

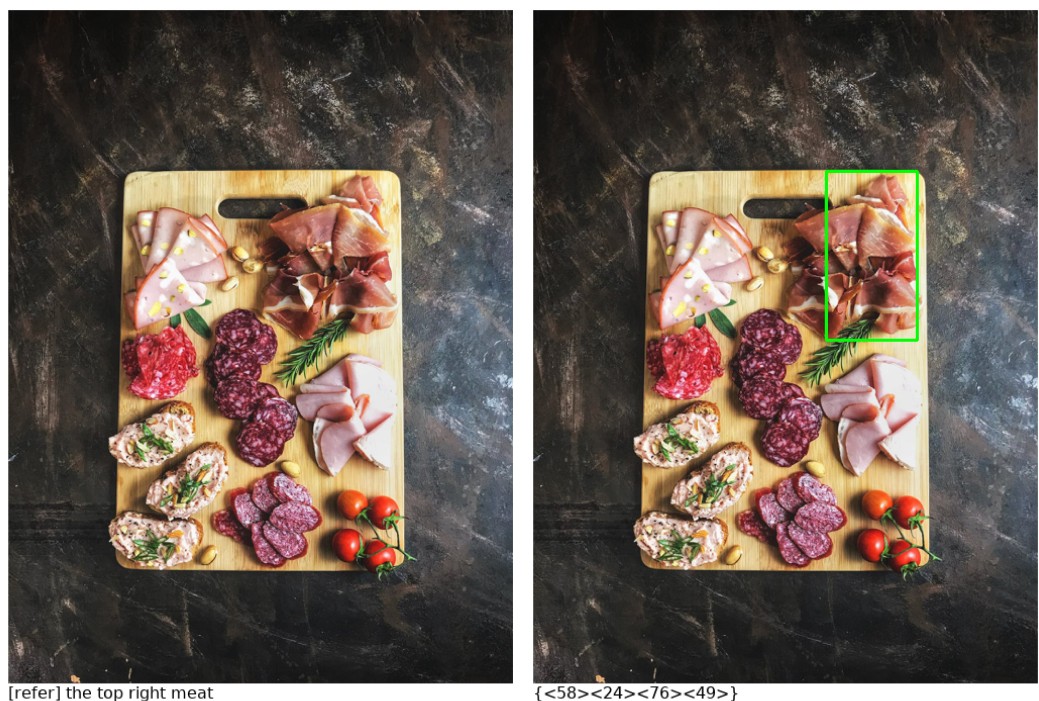

[refer] the top right meat                    {<58><24><76><49>}

Figure 15: Referring expression comprehension example

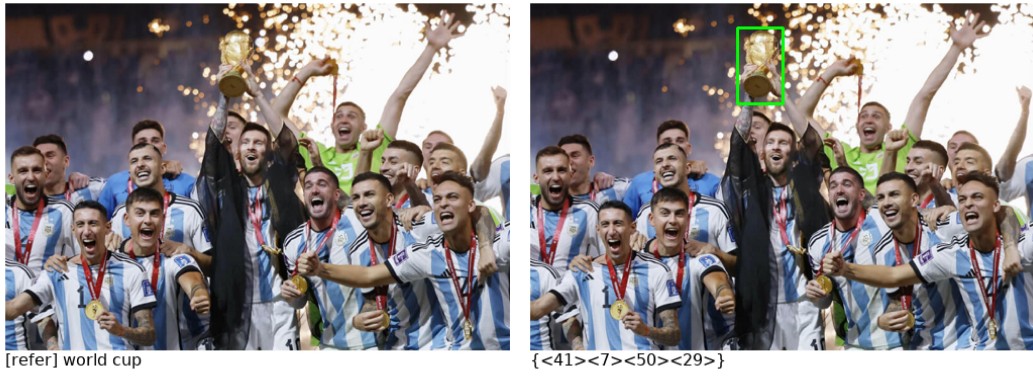

[refer] world cup                    {<41><7><50><29>}

Figure 16: Referring expression comprehension example

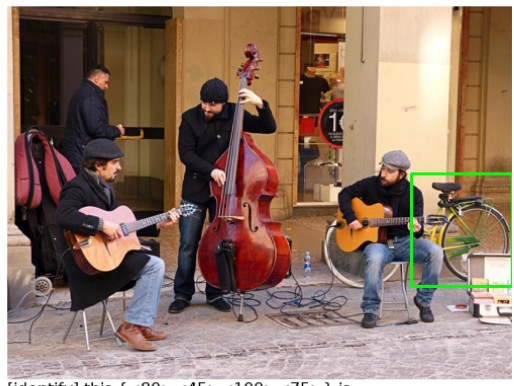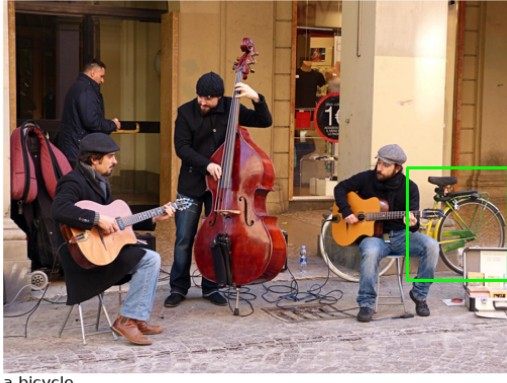

[identify] this {<80><45><100><75>} is    a bicycle

Figure 17: object identification example

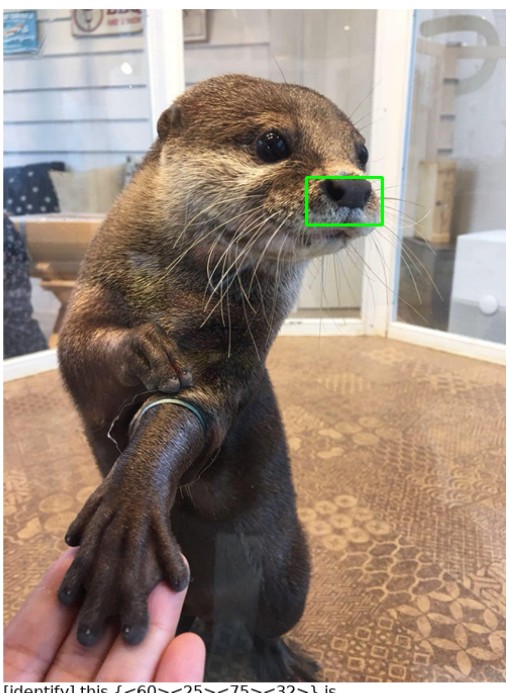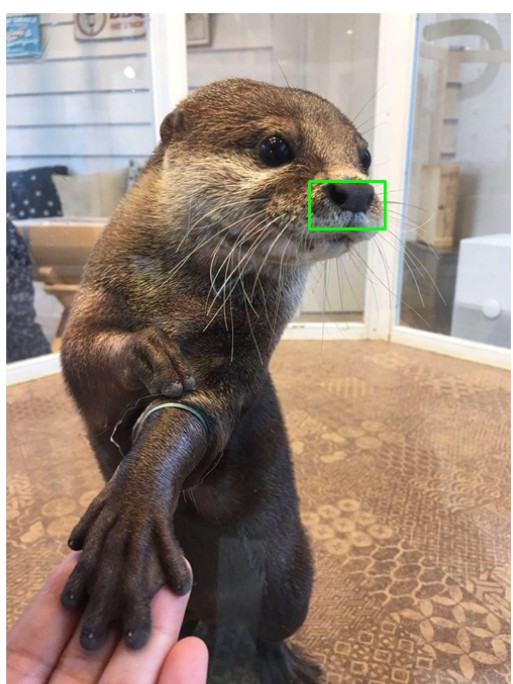

[identify] this {<60><25><75><32>} is    its nose

Figure 18: object identification example

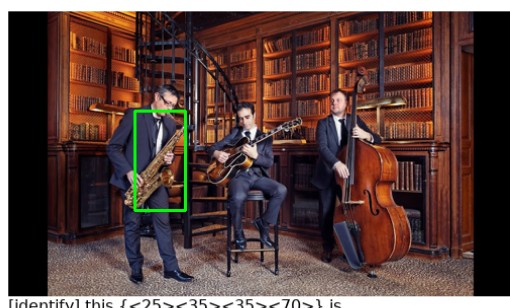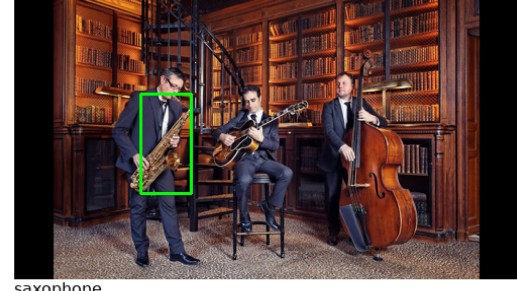

[identify] this {<25><35><35><70>} is    saxophone

Figure 19: object identification example

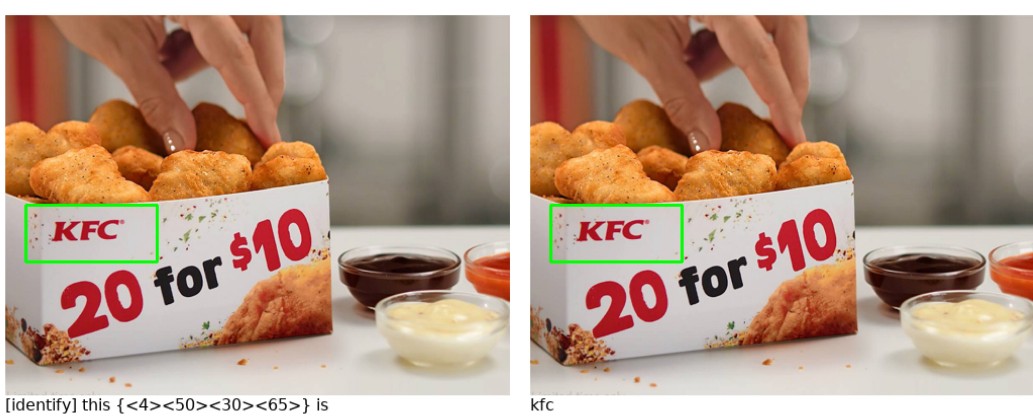

Figure 20: object identification example

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
