# Supplementary for MiniGPT-v2: Large Language Model as a Unified Interface for Vision-Language Multi-task Learning

In the supplementary, we provide more qualitative results that are generated from our model to demonstrate the vision-language multi-tasking capabilities.

## 1 Instruction template for various vision-language tasks

**RefCOCO/RefCOCO+/RefCOCOg:** *[refer] give me the location of question*

**VizWiz:** *[vqa] Based on the image, respond to this question with a short answer: question and reply 'unanswerable' if you could not answer it*

**Hateful Meme:** *[vqa] This is an image with: question written on it. Is it hateful? Answer:*

**VSR:** *[vqa] Based on the image, is this statement true or false? question*

**IconQA, GQA, OKVQA:** *[vqa] Based on the image, respond to this question with a short answer: question*

## 2 Additional Qualitative Results

To study how well our model is able to take visual input and answer questions based on task-oriented identifier, we use our model to perform multiple vision-language tasks including grounded image captioning in Fig. 1, Fig. 2, Fig. 3 and Fig. 4; Object parsing and grounding in Fig. 5, Fig. 6, Fig. 7 and Fig. 8; Referring expression comprehension in Fig. 9, Fig. 10, Fig. 11 and Fig. 12; Object identification in Fig. 13, Fig. 14, Fig. 15 and Fig. 16.

For each task, we share 4 examples for showing the vision-language capabilities of our model. The results in the demo provide direct evidence for the competing visual understanding capabilities of MiniGPT-v2 on multiple vision-language tasks. For example, in the cases of grounded caption, our model is able to give correct grounded image caption with detailed spatial locations of objects. In the cases of identify, the model also generates our expected object names. MiniGPT-v2 can understand the new scenes and follow the question identifier to respond. But we also need to note that our model still has some hallucination e.g., In Fig. 3, several persons are not grounded accurately, and in Fig. 4, there does not exist a vase in the image.

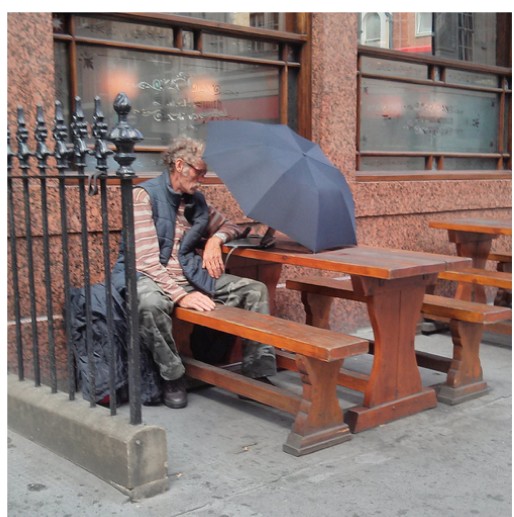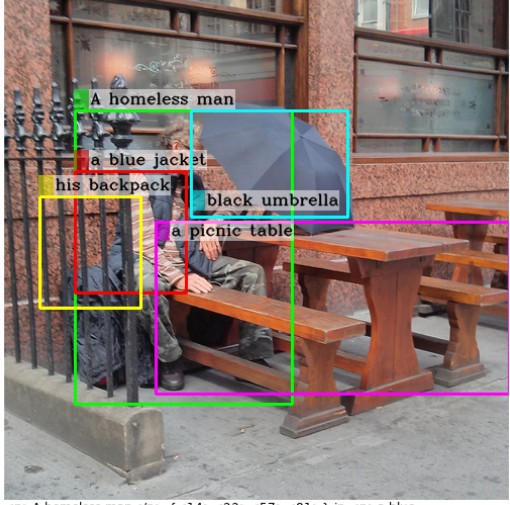

[grounding] please describe this image as detailed as possible

<p>A homeless man</p> {<14><23><57><81>} in <p>a blue jacket</p> {<14><35><36><59>} and <p>black umbrella</p> {<37><23><68><44>} sits at <p>a picnic table</p> {<30><45><100><79>} while <p>his backpack</p> {<7><40><27><62>} is beside him .

Figure 1: Detail grounded image caption example.

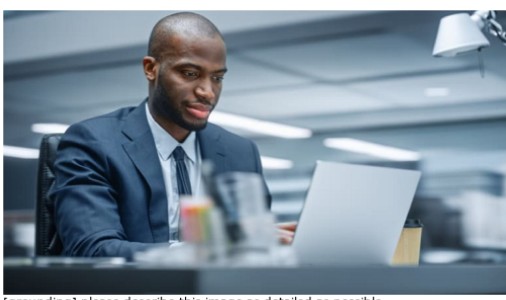

[grounding] please describe this image as detailed as possible

<p>A young African-American business man</p> {<8><1><64><89>} in <p>a blue blazer</p> {<7><31><52><89>} looks up at the camera with <p>a pleased smile</p> {<34><32><45><43>} while working at his desk, using <p>a laptop computer</p>

Figure 2: Detail grounded image caption example

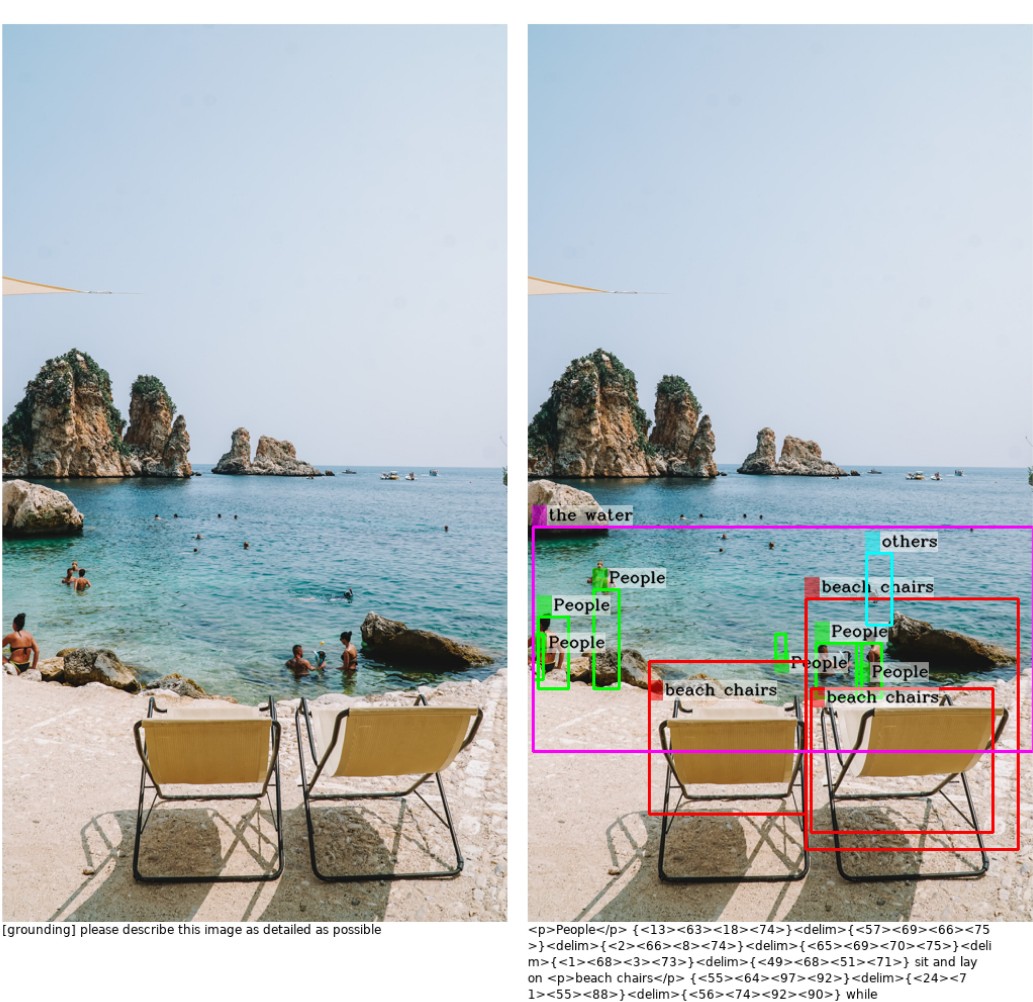

Figure 3: Detail grounded image caption example

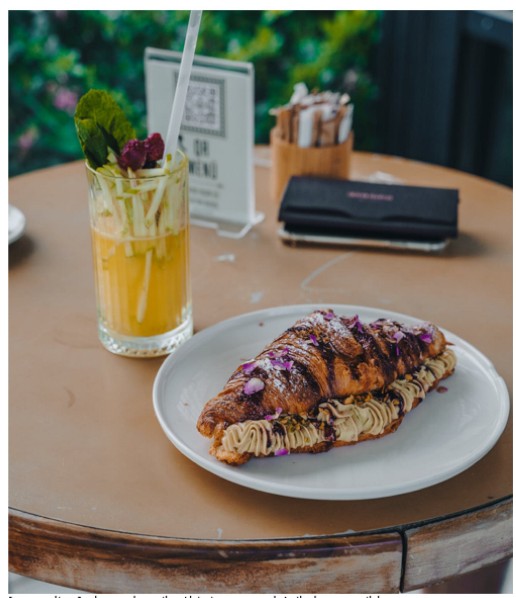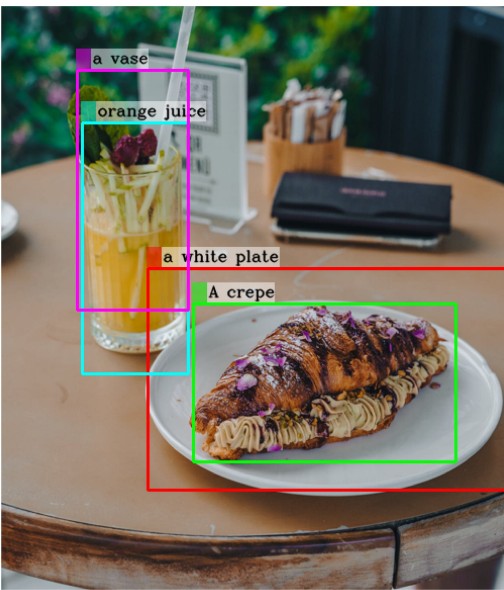

[grounding] please describe this image as detailed as possible

<p>A crepe</p> {<38><51><90><78>} sits on <p>a white plate</p> {<29><45><100><83>} near a cup of <p>orange juice</p> {<16><20><37><63>} and <p>a vase</p> {<15><11><37><52>} with a stem in the background .

Figure 4: Detail grounded image caption example

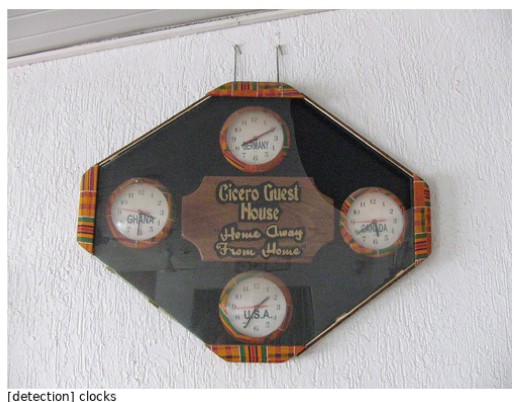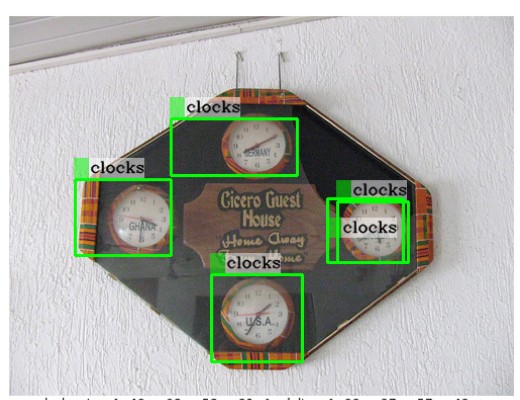

[detection] clocks

<p>clocks</p> {<40><68><58><91>}<delim>{<32><27><57><42>}<delim>{<13><43><32><63>}<delim>{<65><49><79><65>}<delim>{<63><48><78><65>}

Figure 5: Object parsing and grounding example

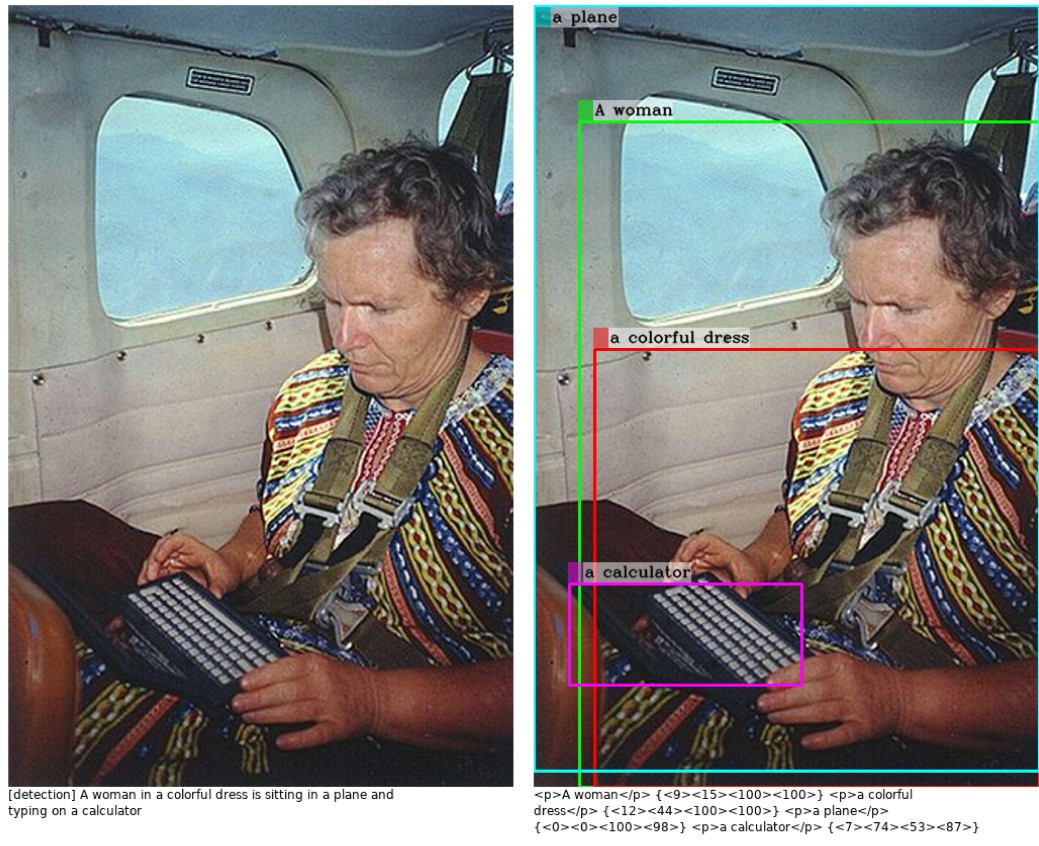

Figure 6: Object parsing and grounding example

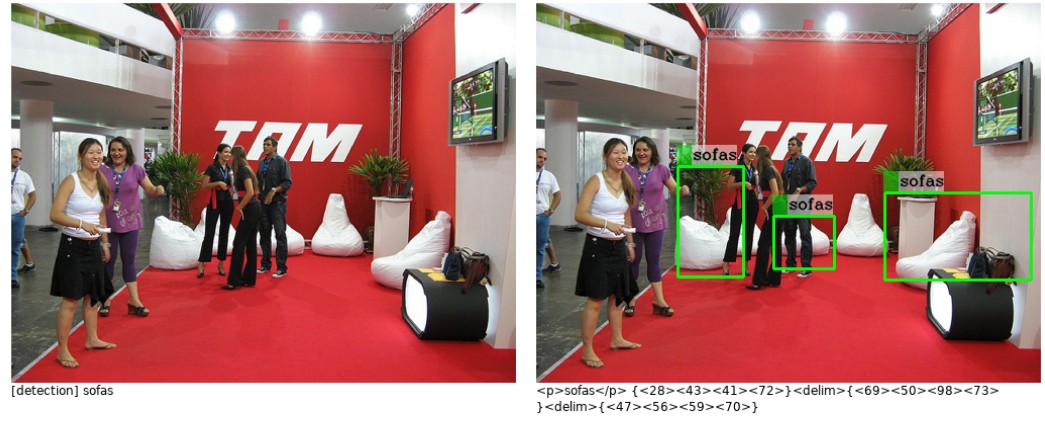

Figure 7: Object parsing and grounding example

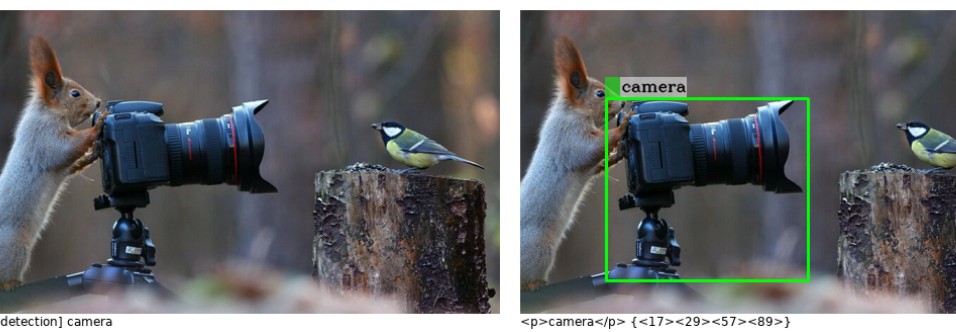

Figure 8: Object parsing and grounding example

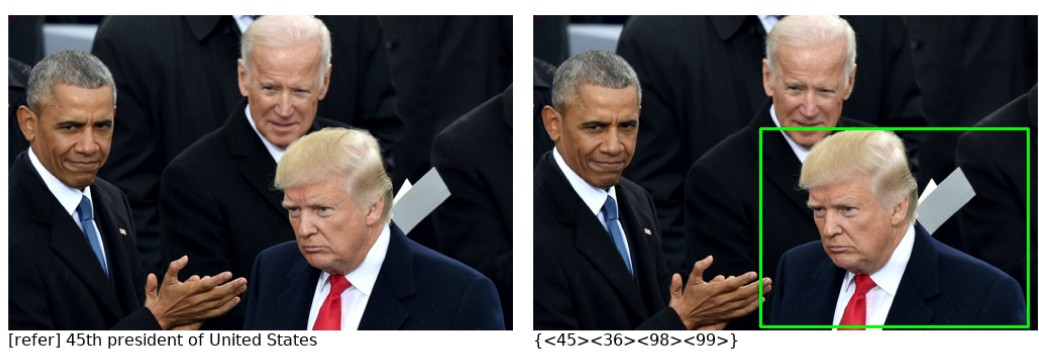

Figure 9: Referring expression comprehension example

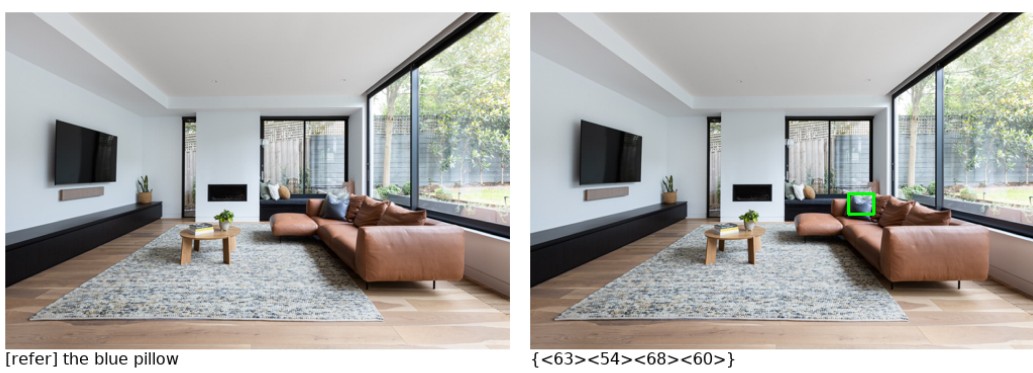

Figure 10: Referring expression comprehension example

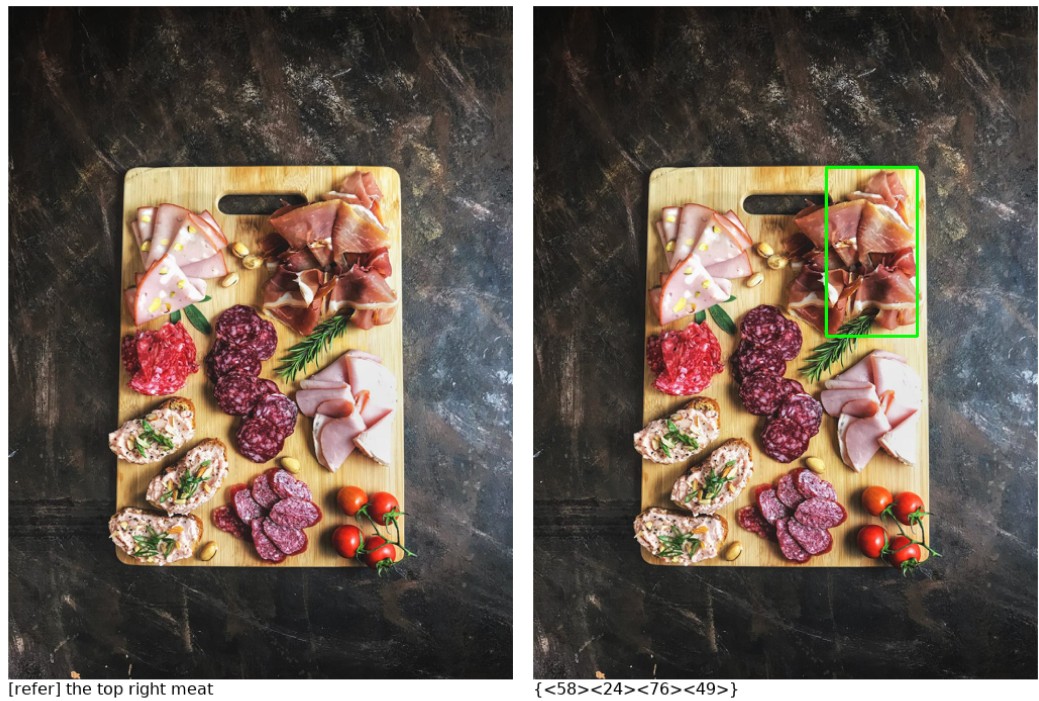

[refer] the top right meat
{<58><24><76><49>}

Figure 11: Referring expression comprehension example

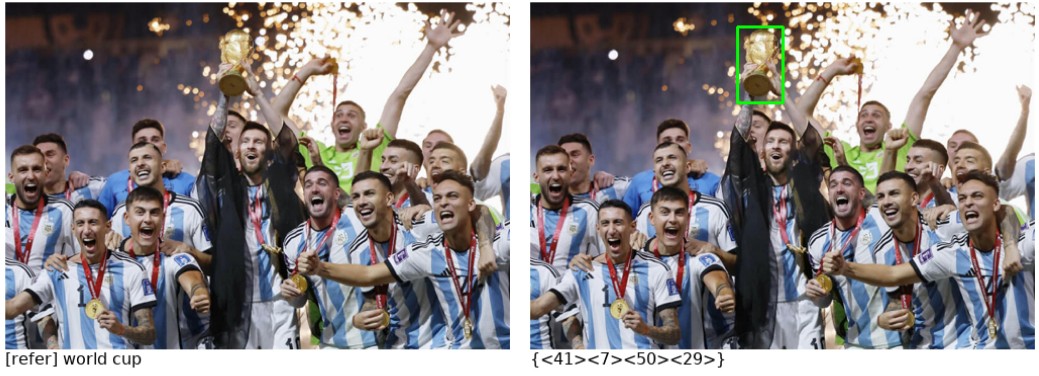

[refer] world cup
{<41><7><50><29>}

Figure 12: Referring expression comprehension example

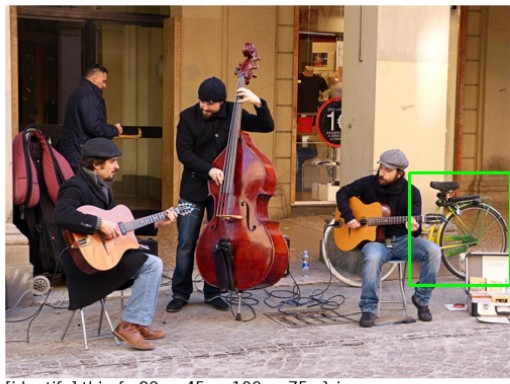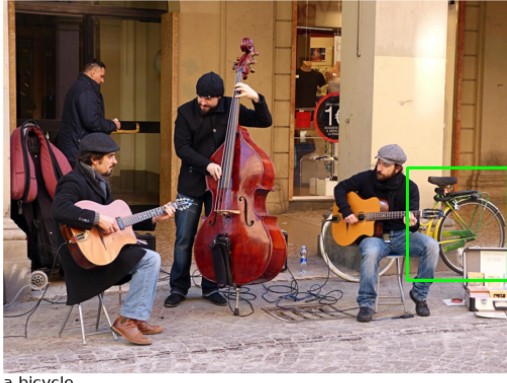

[identify] this {<80><45><100><75>} is          a bicycle

Figure 13: object identification example

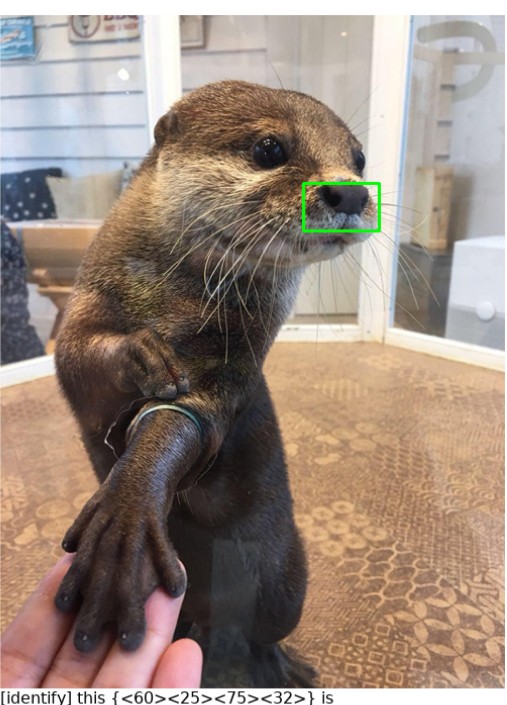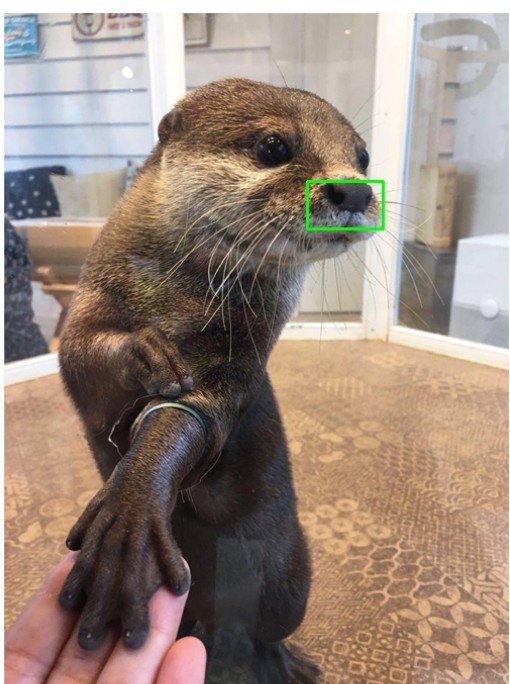

[identify] this {<60><25><75><32>} is          its nose

Figure 14: object identification example

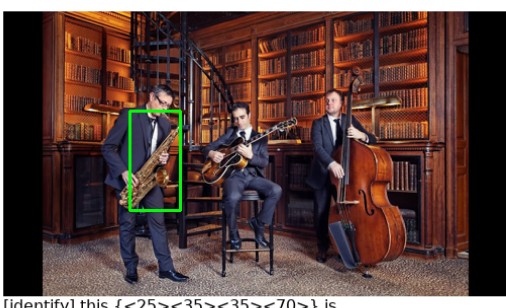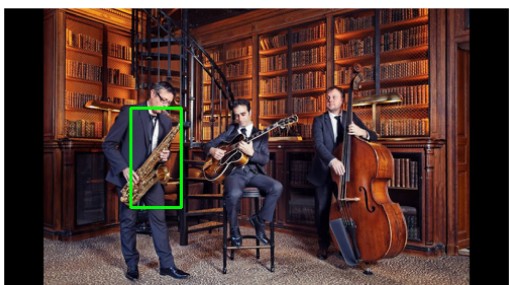

[identify] this {<25><35><35><70>} is          saxophone

Figure 15: object identification example

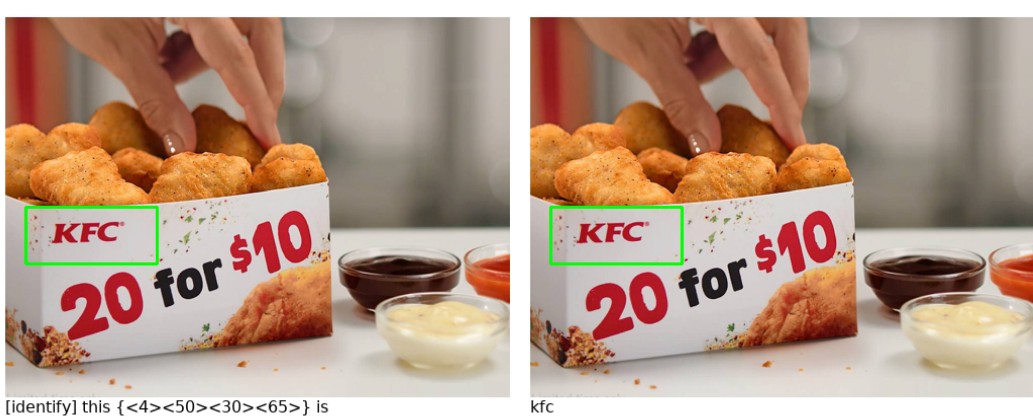

Figure 16: object identification example