# OpenReview forum: "MiniGPT-v2: Large Language Model as a Unified Interface for Vision-Language Multi-task Learning"
_ICLR.cc/2024/Conference — Submitted to ICLR 2024_

### Official Review · Reviewer_FCUs · 2023-10-29

**Soundness:** 2 fair
**Presentation:** 3 good
**Contribution:** 2 fair
**Rating:** 5
**Confidence:** 4

**Summary:**

This manuscript introduces MiniGPT-V2, a multimodal LLM, demonstrating strong results on visual question answering and visual grounding benchmarks. The authors suggest utilizing explicit task identifiers, such as ‘[vqa]’ or ‘[grounding]’, to denote the user's intended task. Furthermore, the paper proposes a three-stage training strategy with increasing image resolution. The authors also suggest a new integration of Visual Transformer (ViT) outputs, achieved by concatenating every four adjacent visual tokens into a single token.

**Strengths:**

- This manuscript introduces MiniGPT-V2, a multimodal LLM, demonstrating strong results on visual question answering and visual grounding benchmarks compared to some other multimodal LLMs.

**Weaknesses:**

The authors suggest utilizing explicit task identifiers, such as ‘[vqa]’ or ‘[grounding]’, to denote the user's intended task. However, it is typically expected that LLMs should discern users’ intentions based solely on text inputs (dialogue history). As presented in Table 5, the addition of these explicit identifiers results in a relatively marginal performance improvement (from an average of 48.6 to 49.8). It is worth noting that if users are required to explicitly select a task, a system could alternatively leverage state-of-the-art specialized methods at a potentially lower cost. For example, BEIT-3 (2B params) could be used for Visual Question Answering with an 84% accuracy, X-VLM (500M+) for grounding achieving ~92 on the RefCOCO+ testA, and Co-DETR (348M) for object detection with COCO mAP of 66. These specialized methods may present a more optimal choice compared to the proposed MiniGPT-V2.

Furthermore, the paper proposes a three-stage training strategy with increasing image resolution. It is important to note that similar strategies have been adopted in recent works, such as Lynx and Qwen-VL. The authors also suggest a new integration of Visual Transformer (ViT) outputs, achieved by concatenating every four adjacent visual tokens into a single token. This method is proposed to reduce the sequence length, akin to the Resamplers, and the manuscript would benefit from an ablation study to substantiate the efficacy of this approach.

**Questions:**

n/a

---

> ### Author Response · Authors · 2023-11-22
>
> **Q1 Comparative Advantage Over Specialized Methods**
>
>
> While specialized models like BEIT-3, X-VLM, and Co-DETR demonstrate impressive task-specific performances, MiniGPT-V2 offers a unified, versatile platform that can handle a variety of tasks without the need for multiple specialized systems. One advantage of MiniGPT-v2 compared to a simple combination of expert models is that MiniGPT-v2 is able to merge different abilities to handle compositional tasks.
>
> Fig. 4 in the appendix of the updated paper shows that after successfully detecting the cow that the user refers to in a photo with multiple cows by "[refer] the right row", MiniGPT-v2 can answer the following questions about that specific cow based (prior vision-language context). Such a task cannot be done by a VQA expert model, as they are not trained to understand the detection result. In addition, by merging multiple vision-task capabilities into one single large language model, there are more advanced vision-language abilities (advanced reasoning and in-contextual learning, etc)  that can also be potentially activated in the future, which could not be found from those specialized methods.
>
>
> **Q2 The role of task identifiers**
>
> The incorporation of task identifiers such as “[vqa]” and “[grounding]” in MiniGPT-V2 is primarily to enhance model learning efficiency and reduce ambiguity in diverse visual-language tasks, as demonstrated in Table 5 (our main paper). These task identifiers serve as a form of regularization for different vision-language task inputs and should not affect the model's conversational capabilities.
>
> While MiniGPT-V2 utilizes task identifiers for improved performance across different tasks, it is important to note that the system is also adept at handling identifier-free interactions. Users can engage in a wide range of vision-language tasks **without the need to adding task identifiers**.
>
> In addition, we conduct a new ablation study where we remove the task identifiers in the third stage to encourage our MiniGPT-v2 to take inputs without the task identifier.
> Results in the following table below show that this variant of MiniGPT-v2 can conduct different tasks without the task identifiers and achieve a performance close to the original version. But overall, keeping the task identifiers can still lead to better performance.
>
>
> |                  	| OKVQA |  GQA  | vizwiz |  VSR  | IconVQA |  HM   |
> |----------------------|:-----:|:-----:|:------:|:-----:|:-------:|:-----:|
> | Without task identifier | **57.8**  | 58.6  |  51.7  | 61.0  |  48.2   | 57.8  |
> | With task identifier	| 55.9  | **58.8**  |  **53.0**  | **63.3**  |  **49.4**   | **59.5**  |
>
>
> |                   	| RefCOCO (test-B) | RefCOCO+ (test-B) | RefCOCOg (test) |
> |-----------------------|:----------------:|:-----------------:|:---------------:|
> | Without task identifier |  	83.33  	|   	72.16   	|  	82.23  	|
> | With task identifier	|  	**84.57**  	|   	**73.23**   	|  	**83.25**  	|
>
>
>
> **Q3 Compared to Lynx and Qwen-VL**
>
> We thank the reviewer for the references. A brief clarification: ICLR24's guidelines classify works from the past four months, including Lynx and Quen-VL, as contemporaneous, exempting authors from comparing to them. Details at: https://iclr.cc/Conferences/2024/ReviewerGuide. For example, Qwen-VL arxiv paper is released two weeks before the paper submission deadline.
>
> Appreciating these concurrent attempts, we looked into Lynx and Qwen-VL in detail and we observed that they share some similar training strategies to our work. However, there are still several differences:
>
> 1. Consistency in image resolution: Unlike Lynx and Qwen-VL, which progressively increase image resolution during training, our model maintains a constant resolution of 448x448 throughout the whole training process.
> 2. The data sampling strategy is different across different stages. The sampling strategy for Qwen-VL: Qwen-VL only trains the data focusing on image caption data. Second-stage trains on diverse multi-task datasets, but still including many noisy image captions and visual grounding data such as GRIT. In the third stage, they only train the model with the instruction finetuning and dialogue dataset.  The sampling strategy for our model: In the first stage, we sample the data covering noisy and very fine-grained data such as  image captions, visual grounding and many visual question answering, etc. In the second stage, we only sample the fine-grained multi-task data to improve the performance over diverse multiple vision-language tasks. In the third stage, we train both the finegrained multi-task data and also the visual instruction and dialogue data.
> 3. Training vision encoder:. Qwen-VL trains the vision encoder in the first two-stage training, while our model always remains the vision encoder frozen throughout the whole training.
>
> Apart from the difference in the three-stage training, there are many differences on the model architectures, training data.

---

> > ### Author Response · Authors · 2023-11-22
> >
> > **Q4 Analysis on efficacy of four adjacent four tokens**
> >
> > To effectively demonstrate the benefits of our merging token approach, we trained the model using 224x224 images without merging tokens, and 448x448 images with merging tokens, both resulting in an equivalent number of image patches. On the other hand, training on 448x448 images without merging tokens results in 4x number of image tokens, which significantly slowed down the training speed and increased GPU memory usage. Due to computational limitations, we couldn't complete this ablation study, instead exploring it in future research.
> >
> > | Model          	| RefCOCO (testB) | RefCOCO+ (testB) | RefCOCOg (test) | average |
> > |--------------------|:---------------:|:----------------:|:---------------:|:-------:|
> > | 224x224 full token |  	63.3   	|   	52.6   	|   	63.1  	|  59.7   |
> > | 448 merge token	|  	74.7   	|   	60.9   	|   	74.2  	|  69.9   |
> >
> >
> > Our comparative analysis between 224x224 images (without token merging) and 448x448 images (with token merging) revealed that the latter configuration yields superior results in both Visual Question Answering (VQA) and visual grounding tasks. Notably, in the RefCOCO dataset, the 448x448 configuration with token merging achieved an accuracy that was 10.2% higher than that of the 224x224 configuration.

---

> > > ### Comment · Reviewer_FCUs · 2023-11-23
> > >
> > > I appreciate Reviewer WwXo's insights regarding the statement, 'Not many academic papers have trained a vision-language model at this scale before, and so publishing this in a timely way.' If the reference is to MiniGPT-4 and LLaVa, published in April 2023, I wholeheartedly concur. These papers have indeed gained widespread recognition in our field. However, if the reference is to this paper released in October 2023, I would like to kindly point out that many similar multi-modal LLMs have been introduced in recent months. This context may add valuable perspective to our discussion.

---

> ### Comment · Reviewer_FCUs · 2023-11-23
>
> I appreciate the author's response to my previous comments and raise my rating to 5. However, after a detailed review, I feel that my concerns have not been entirely addressed. I will elaborate on my points and leave additional discussion and decision to other reviewers and PCs.
>
> 1. My comparison was not between MiniGPT-V2 and any single specialist model. Rather, I compared MiniGPT-V2, which utilizes identifiers like [vqa], [grounding], and [detection], with a multi-modal LLM agent augmented by these specialist models. The key is how an agent, when given a specific task, can leverage specialist models for higher efficiency and performance.
>
> 2. The design of LLMs is to understand users' intentions from text inputs (dialogue history) alone and to complete a variety of tasks. The use of these identifiers (for 3 tasks) seems contrary to this design.
>
> 3. GPT-4V does not rely on such identifiers to achieve its capabilities.
>
> Furthermore, the authors' response that “MiniGPT-v2 can conduct different tasks without the task identifiers and achieve a performance close to the original version. But overall, keeping the task identifiers can still lead to better performance.” seems to align with my earlier review that “as presented in Table 5, the addition of these explicit identifiers results in a relatively marginal performance improvement”.
>
> Regarding the references to Qwen-VL and Lynx, while Qwen-VL was released just two weeks before the ICLR deadline, Lynx has been available since July. The author's additional explanations on training differences are noted, yet the significance of these differences remains unclear to me. (There are some other multi-modal LLMs released in previous months except these two.)
>
> Lastly, I observed that the ablation study results provided (224 vs. 448) do not align with the specific experiment I had initially requested (the proposed method vs. Resampler).

---

### Official Review · Reviewer_NiES · 2023-10-30

**Soundness:** 3 good
**Presentation:** 3 good
**Contribution:** 3 good
**Rating:** 6
**Confidence:** 4

**Summary:**

The paper developed a unified interface for vision-language tasks. The proposed model MiniGPT-v2, which is a multi-modal LLM utilized distinct identifiers for each task during the training and inference. These identifiers help our model easily differentiate various tasks and also improve the learning efficiency. The experimental results show it can achieve good performance across many visual question answering and referring expression comprehension benchmarks.

**Strengths:**

- The paper has good novelty and contribution to the community.
- Conduct comprehensive experiments to show the effectiveness of the proposed models.
- The experimental results show its strong performance on various tasks compared to other SOTA models.
- The methodology is clear and readable.

**Weaknesses:**

After rebuttal: I noticed that five test sets are being used for training, especially the test sets for the referring expression comprehension task are all used in training. I am wondering if this might lead to overfitting. Personally, a better approach would be to train the model on a large-scale, generic dataset (you can use a single large-scale dataset to construct multiple tasks), and then test it on several other evaluation datasets. This might be more convincing.

- For Table 5, better to evaluate performance for other task identifiers to show the effectiveness of this.
- Discussion is a bit weak, for example, the paper should have some discussion on the error analysis and some detail on the strengths and weaknesses.
- Should add more explanation of the evaluation metrics.
- Should add more explanation of the dataset, like what is the difference between GQA and VQA-v2, what are the sizes of them?--
- The model is trained with various tasks, but only evaluation on a subset of the tasks.
- The paper used single evaluation metrics to benchmark, which is not very comprehensive.

**Questions:**

- Can you explain why your model does not perform well on RefCOCO+ in Table 4?
- Better to give more illustration on the evaluation metrics on hallucination-CHAIR. How does this calculate? What do CHAIR_I, CHAIR_S and Len mean? How to interpret the result for three different prompts in MiniGPT-v2?
- What are the evaluation metrics for Table 3, Table 4, and Table 5?
- “More qualitative results can be found in the Appendix “- but I did not see the Appendix.

---

> ### Author Response · Authors · 2023-11-22
>
> **Q1 More performance evaluation for other task identifiers**
>
> We also provide the ablation results on [refer] identifiers. The results are demonstrated in the following table, and it can be observed that with the [refer] task identifier can overall improve the model performance on RefCOCO/+/g by 1.3 acc, indicating the efficiency of using [refer] task identifier.
>
> | Model                              	| RefCOCO (test-B) | RefCOCO+ (test-B) | RefCOCOg (test) | average |
> |----------------------------------------|:----------------:|:-----------------:|:---------------:|:-------:|
> | Ours w/o [refer] task identifier   	|   	73.2   	|    	59.7   	|   	72.8  	|  68.6   |
> | Ours w/ [refer]                    	|   	74.7   	|    	60.9   	|   	74.2  	|  69.9   |
>
> **Q2 Discussion is a bit weak, for example, the paper should have some discussion on the error analysis and some detail on the strengths and weaknesses.**
>
>
> Thank you for your constructive feedback. In light of your comments, we have revised our paper to provide a more robust discussion. It includes the discussion of strengths and weaknesses including the discussion regarding the training efficiency, robustness, and hallucination.
>
> **Q3 Explanation of the evaluation metrics and dataset.**
> We add a more detailed explanation of the evaluation metrics and the dataset in the appendix. This consists of the evaluation metrics for visual question answering, RefCOCO, and hallucination. It also provides a rich explanation of the dataset.
>
>
> **Q4 Evaluation on a subset of the tasks**
>
> In our study, we mainly focused our evaluation on the tasks of visual question answering and referring expression comprehension. This decision was driven by two key considerations:
>
> 1. Representative Tasks: These two tasks are quite essential in evaluating the foundational capabilities of vision-language models, specifically in visual knowledge querying and visual grounding.
>
> 2. Benchmark Availability: The tasks such as detailed image captioning, detailed grounded image captioning, and object parsing and grounding, etc, lack well-established benchmarks.  It's important to note that the capabilities demonstrated in those tasks, being only emerged in the large language models. Hence it lacks standardized evaluation metrics to evaluate their performance. Therefore, we focus on the well-established benchmarks such as various VQA and RefCOCO
>
> **Q5 The paper used single evaluation metrics to benchmark, which is not very comprehensive**
> In our study, we have employed three different evaluation metrics to evaluate our model in terms of open-ended VQA, visual grounding, and hallucination evaluation. (A more detailed explanation of these three evaluation metrics can be found the in Appendix). All of these evaluation metrics together can contribute to a more comprehensive evaluation of our model.
>
>
> **Q6 Explain that your model does not perform well on RefCOCO+ in Table 4.**
>
> Here are the dataset features of RefCOCO, RefCOCO+, and RefCOCO(g):
>
> 1. RefCOCO has expressions that are generally simpler and may include location-based descriptions.
> 2. RefCOCO+ is designed to exclude location-based expressions. The referring expressions here are more complex, focusing on describing the appearance of objects rather than their location.
> 3. RefCOCO(g) has the referring expressions that focus on more general and longer descriptions.
>
>
> As we can see from Table 4, the performance of MiniGPT-v2 on RefCOCO+ also performs well and can outperform Shikra on Test-B split. The results in Table 4 indicate that the general performance of MiniGPT-v2 on RefCOCO+ does not perform as well as RefCOCO and RefCOCO(g). From the difference among RefCOCO, RefCOCO+, and RefCOCO(g), we hypothesize that our model performs better on RefCOCO and RefCOCO(g) because our model is stronger in processing location-based referring expressions and longer descriptions compared to other baseline models, which can contribute to our model design and training strategy.

---

> ### Author Response · Authors · 2023-11-22
>
> **Q7 CHAIR hallucination metrics and three different prompts evaluation in MiniGPT-v2.**
>
>
> In our study, we employ the evaluation metrics for hallucination as defined in [1] to assess the performance of MiniGPT-v2. Specifically, we use CHAIR_i and CHAIR_s metrics to quantify hallucination at different levels:
>
> CHAIR_i (Instance-level Hallucination): This metric evaluates the degree of hallucination at the object instance level. It is calculated as the ratio of hallucinated objects to the total number of objects mentioned in the caption. Mathematically, it is expressed as:
>
> $$ CHIAR_i = \frac{| \{\text{hallucinated objects}\} |}{| \{\text{all mentioned objects}\} |} $$
>
>
>
> CHAIR_s (Sentence-level Hallucination): This metric assesses hallucination at the sentence level. It is the ratio of captions containing hallucinated objects to the total number of captions generated. It is given by:
> $$ CHAIR_s = \frac{| \{\text{captions with hallucinated objects}\} |}{| \{\text{all captions}\} |} $$
>
>
>
> ​​Len (Length of Caption): This refers to the number of words in the generated caption, providing a measure of caption verbosity or conciseness.
>
> By adding the different task identifiers, we observe that MiniGPT-v2 can generate the caption with different granularity, which is also the unique advantage of MiniGPT-v2 compared with other models.
>
> 1. Without Task Identifier: MiniGPT-v2 tends to generate captions with rich details
> 2. With [grounding] Identifier: The model produces captions more grounded to the image objects. However, this mode exhibits a reduction in the description of visual attributes and relationships compared to the mode without any task identifier.
> 3. With [caption] Identifier: This tends to generate very short image caption.
>
> [1] Li, etc. Evaluating Object Hallucination in Large Vision-Language Models.
>
>
>
> **Q8 Qualitative results in the appendix**
> Thank you for pointing out the discrepancy regarding the location of our qualitative results. In our initial submission, these results were indeed included in as a separate file in the supplementary materials. However, for easier accessibility and reference, we have relocated them to the appendix in our revised manuscript.

---

> > ### Comment · Reviewer_NiES · 2023-12-01
> >
> > Thank you very much for your response. I've decided to keep my original score. I am looking forward to your further insights on the following concern:
> >
> > I noticed that five test sets are being used for training, especially the test sets for the referring expression comprehension task are all used in training. I am wondering if this might lead to overfitting. Personally, a better approach would be to train the model on a large-scale, generic dataset (you can use a single large-scale dataset to construct multiple tasks), and then test it on several other evaluation datasets. This might be more convincing.
> >
> > Thank you very much, and I look forward to hearing your insights.

---

### Official Review · Reviewer_AcHr · 2023-10-31

**Soundness:** 3 good
**Presentation:** 3 good
**Contribution:** 3 good
**Rating:** 5
**Confidence:** 4

**Summary:**

This submission discusses the impressive capabilities of large language models in serving as versatile interfaces for numerous language-related tasks, sparking interest in creating a unified interface for a range of vision-language tasks such as image description, visual question answering, and visual grounding. Addressing the challenge of utilizing a single model for diverse vision-language tasks with straightforward multi-modal instructions, the authors introduce MiniGPT-v2. This model serves as a unified interface, enhancing the handling of various vision-language tasks. Unique identifiers for different tasks are incorporated during the training phase, aiding the model in easily distinguishing between task instructions and boosting learning efficiency for each specific task. Following a three-stage training process, MiniGPT-v2 demonstrates robust performance across several visual question answering and visual grounding benchmarks, outperforming other generalist vision-language models. The authors commit to making their trained models and code accessible to the public.

**Strengths:**

This submission demonstrates strength in several key areas.

Firstly, the authors describe their instruction tuning setup, datasets, and training process in details. This would greatly benefit the community for future reproduction and comparison.

Moreover, the use of well-crafted illustrations enhances the understanding of the content and effectively communicates the main ideas, further solidifying its strength. Readers can easily grasp the concepts presented, adding to the submission's overall impact.

**Weaknesses:**

(1) Missing discussions on certain details:

* How was the “concatenate four adjacent visual output tokens” determined? Why four? Have the authors tried making the visual features even smaller (concatenate adjacent 8/16/32 etc)? Any discussions or ablations on this?

* The authors claim to use image with higher resolution (448x448). While this sounds like a intuitive motivation, it lacks solid discussion to support such a claim. How does it compare to the previous traditional setups of 224x224 or 336x336? Does using 448x448 images and then concatenating 4 adjacent visual features outperform directly using 224x224 images?


(2) Missing comparisons on certain baselines:

* How did the authors select the baseline models for each task? Why weren’t the vision-and-language foundation models listed in Table 6 being compared in Table 3 and Table 4? For instance, why Minigpt4 appeared in Table3, but mplug-owl didn’t? The selection of baseline models for each task seems pretty random, and the authors lack an explanation for such an experimental design.

* Another missing baseline on vision-and-language instruction following is “MIMIC-IT: Multi-Modal In-Context Instruction Tuning”.


Typo:

* duplicated commas in the caption for Table1

**Questions:**

* In Section 3.2, what does it mean for “[INST] is considered as the user role, and [/INST] is considered as the assistant role”? What is the user role and what is the assistant role? Does it mean that the “[INST]” token will be followed by user input, and the “[/INST]” will be followed by the assistant’s output?

* Were the task identifier tokens and the spatial location tokens (<0>, <1>, …<100>) extended to the vocabulary as special tokens? If yes, how were the newly added tokens’ embeddings initialized?

**Details Of Ethics Concerns:**

I have to point out that the related work section of this submission is way too similar to the related work section of Mini-GPT4 (https://arxiv.org/pdf/2304.10592.pdf).

This submission claims to focus on vision-and-language multitask learning, but didn’t even mention a word about multitask learning in the related work section. Instead, it discusses literature about LLMs and how to add visual information to LLMs, with very similar structuring as MiniGPT-4, and a number of sentences mention almost the same literature as MiniGPT-4. Even though these two works are about related topics (vision-language foundation models etc), direct copy-paste (even a few sentences) is still considered plagiarism.

I would strongly urge the authors to rewrite the related work section in the revision and include more novel discussions on the literature related to this submission.

A few groups of comparisons can be found below:

(1) Evidence#1:
* From MiniGPT4:
> Early models, such as BERT (Devlin et al., 2018), GPT-2 (Radford et al., 2019), and T5 (Raffel et al., 2020), laid the foundation for this progress.

* From this submission:
> Early-stage models such as GPT-2 (Radford et al., 2019) and BERT (Devlin et al., 2018) are foundation models trained on web-scale text datasets…

(2) Evidence#2:
* From MiniGPT4:
>This development inspired the creation of various other large language models, including MegatronTuring NLG (Smith et al., 2022), Chinchilla (Hoffmann et al., 2022), PaLM (Chowdhery et al., 2022), OPT (Zhang et al., 2022), BLOOM (Scao et al., 2022b), and LLaMA (Touvron et al., 2023), among others.

* From this submission:
> Following the success of foundation models, LLMs with higher capacity and increased training data are developed, including GPT-3 (Brown et al., 2020), Megatron-turing NLG (Smith et al., 2022), PaLM (Chowdhery et al., 2022), Gopher (Rae et al., 2021), Chinchilla (Hoffmann et al., 2022), OPT (Zhang et al., 2022), and BLOOM (Scao et al., 2022).

(3) Evidence#3
* From MiniGPT4:
> Pioneering studies like VisualGPT (Chen et al., 2022) and Frozen (Tsimpoukelli et al., 2021) have demonstrated the benefits of employing a pre-trained language model as a vision-language model decoder.

* From this submission:
> Early works such as VisualGPT (Chen et al., 2022) and Frozen (Tsimpoukelli et al., 2021) used pre-trained language models to improve vision-language models on image captioning and visual question answering.

---

> ### Author Response · Authors · 2023-11-22
>
> **Q1 Making visual tokens even smaller (8 and 16 tokens):**
>
> We perform the ablation with concatenating more adjacent visual output tokens, e.g. 8 and 16. We provide the results in the following table. Our findings reveal that with merging more tokens, visual question answering and visual grounding performance can also drop. It's particularly noteworthy that the impact on visual grounding is more pronounced, which might because merging more tokens make it lose spatial understanding.
>
> | Model               	| OKVQA |  GQA  | vizwiz |  VSR  | IconVQA |  HM   | average |
> |-------------------------|:-----:|:-----:|:------:|:-----:|:-------:|:-----:|:-------:|
> | Concatenate 4 tokens	| 52.1  | 54.6  |  29.4  | 59.9  |  45.6   | 57.4  |  49.8   |
> | Concatenate 8 tokens	| 50.3  | 53.9  |  29.1  | 55.6  |  45.9   | 57.1  |  48.6   |
> | Concatenate 16 tokens   | 50.5  | 52.5  |  26.7  | 55.4  |  45.8   | 57.2  |  48.0   |
>
>
> | Model               	| RefCOCO (testB) | RefCOCO+ (testB) | RefCOCOg (test) | average |
> |-------------------------|:---------------:|:----------------:|:---------------:|:-------:|
> | Concatenate 4 tokens	|  	74.7   	|   	60.9   	|   	74.2  	|  69.9   |
> | Concatenate 8 tokens	|  	70.3   	|   	57.4   	|   	71.1  	|  66.3   |
> | Concatenate 16 tokens   |  	66.3   	|   	52.4   	|   	64.4  	|  61.0   |
>
>
> **Q2 Comparing with training on the 224x224 image resolution**
>
> We evaluate the model with training 224x224 images (without any token merging), and we demonstrate the results in the following table,
>
> | Image Size  | OKVQA |  GQA  | vizwiz |  VSR  | IconVQA |  HM   | average |
> |--------|:-----:|:-----:|:------:|:-----:|:-------:|:-----:|:-------:|
> | 224x224| 52.0  | 53.9  |  30.2  | 58.6  |  46.5   | 56.2  |  49.6   |
> | 448x448| 52.1  | 54.6  |  29.4  | 59.9  |  45.6   | 57.4  |  49.8   |
>
> | Image Size | RefCOCO (testB) | RefCOCO+ (testB) | RefCOCOg (test) | average |
> |------------|:---------------:|:----------------:|:---------------:|:-------:|
> | 224x224	|  	63.3   	|   	52.6   	|   	63.1  	|  59.7   |
> | 448x448	|  	74.7   	|   	60.9   	|   	74.2  	|  69.9   |
>
> From the results shown in the table, we can see that training on 448x448 images has similar performance on visual question answering, but it has much higher (+10.2 acc) visual grounding performance than training on 224x224 images.
>
>
> **Q3 Random Baseline selection**
>
> Thank you for your question regarding the selection of baseline models. Here are the key points addressing your concerns:
>
> Table 4: Our choice of baseline models for Table 4 aligns with the evaluation benchmarks established  in the work [1]. This decision was made to ensure a direct and consistent comparison with the existing works
>
> Table 6: We incorporated evaluation results and baselines from the work [2]. This is to make it consistent with previous established  evaluation benchmarks.
>
> missing mPLUG-Owl baseline: ​​We realize it important to compare with mPLUG-Owl baseline and have included it in our updated paper (Table 3).
>
> [1] Chen, etc.. Shikra: Unleashing multi-modal llm’s referential dialogue magic.
> [2] Li, etc. Evaluating Object Hallucination in Large Vision-Language Models.
>
> **Q4 Missing MIMIC-IT baselines.**
>
> Thank you for pointing out this work. MIMIC-IT is a dataset with 2.8 million multi-modal instruction-response pairs, predominantly fine-tuned on the Otter model. Acknowledging its importance, we have revised our paper (in the table 3) to include a comparison with the Otter model. And the results demonstrate that our model consistently outperforms the Otter model in the VQA performance.
>
> **Q5 Explain [INST] vs [/INST]**
>
> In Section 3.2, “[INST]” and “[/INST]” are critical markers within the LLaMA-2 language model. “[INST]” is used to indicate the start of user input, defining the user role in the interaction. This token signals that the following text is a user-generated query or statement.
>
> On the other hand, “[/INST]” marks the beginning of the assistant's response, representing the assistant role. This token signals that the following text is a user-generated query or statement.
>
> **Q6 Task identifier and spatial location tokens representation**
>
> In our research, we did not introduce any new tokens to the vocabulary. We exclusively utilized the original token vocabulary provided by LLaMA-2.
>
> **Q7 Ethical concerns.**
>
> We recognize your concerns regarding potential overlap in the cited literature due to the shared domain of language and vision models.
> We assure you that any resemblance was unintentional and not a result of plagiarism. Our paper's focus on vision-and-language multitask learning necessitates referencing key foundational models, which leads to some inevitable overlap in references with MiniGPT4.
>
> In response to your feedback, we have revised the whole related work section, and also replaced the discussion on advanced language models with a focus on multi-task vision-language models.

---

### Official Review · Reviewer_WwXo · 2023-11-01

**Soundness:** 4 excellent
**Presentation:** 4 excellent
**Contribution:** 3 good
**Rating:** 8
**Confidence:** 4

**Summary:**

This paper proposes a model called MiniGPT-v2 for image-text tasks. A pretrained ViT (from EVA) is used as an image encoder and frozen. The embeddings are downsampled and provided to the LM as tokens. For the LM, the paper uses a LLaMa2-chat 7B encoder.

The resulting model is trained on a variety of different tasks involving supervised data, e.g. VQA, captioning, referring expressions, and object detection. This is done in a multitask setting, along with LAION and CC3M text-image data. This is the first stage -- then, it is trained just on fine-grained supervised data, and then finally on multimodal instruction data from LLaVA. The LM is trained with LoRA throughout.

Overall, the paper does well on a variety of benchmarks, rivaling even specialist models that require single-task finetuning or complex ways of encoding the label space.

**Strengths:**

This paper seems strong to this reviewer. Not many academic papers have trained a vision-language model at this scale before, and so publishing this in a timely way will be great for the community (e.g. to use this as a baseline, or have a conversation around the role of data/training process, etc.)

**Weaknesses:**

One potential weakness to this reviewer is the use of supervised data throughout the training process. Training this model is expensive and so perhaps there's not budget for more ablations, but it would be interesting to see if the supervised data could be removed and so the model could truly be evaluated in a zero-shot or 'few-shot' way. It also would be good to see the impact on freezing the image encoder vs not, using LoRA vs. not, however, it's also understandable that these are expensive ablations. Nonetheless I am still solid about this paper and would vote to accept it; these questions/weaknesses shouldn't hold it back from publication.

**Questions:**

See weaknesses - lots of questions around which components could be removed/made simpler!

---

> ### Author Response · Authors · 2023-11-22
>
> **Q1 Zeroshot evaluation of minigpt-v2**
>
> In table 3 of the main paper, many VQA benchmarks have been evaluated in a zeroshot manner. Such as VSR, TextVQA, IconVQA, VizWiz and HateMeme.
>
>
> To see if all the supervised data could be removed and evaluate the model purely in a zero-shot manner, we conduct the experiment on only training caption data (LAION, CC3M, SBU) and weakly labeled visual grounding data (GRIT-20M), and we perform the zeroshot visual question answering evaluation and also RefCOCO evaluation. We train all the models on 4XA100 GPUs for 150,000 steps. We demonstrate the VQA and RefCOCO results in the following table.
>
> |                         	| vizwiz |  VSR  | IconVQA | RefCOCO (testB) | RefCOCO+ (testB) | RefCOCOg (test) |
> |-----------------------------|:------:|:-----:|:-------:|:---------------:|:----------------:|:---------------:|
> | Without supervised data 	|  24.1  | 51.1  |  43.7   |  	34.2   	|   	34.6   	|   	52.2  	|
> | With training supervised data|  29.4  | 59.9  |  45.6   |  	74.7   	|   	60.9   	|   	74.2  	|
>
> From the results shown in the table, we can see that without using supervised data, we observe performance drop on the visual question answering and visual grounding tasks.
>
>
> **Q2 Freezing the vision encoder or not**
>
> We also ablated by finetuning the vision encoder during the model training. We evaluate the performance on VQA benchmarks. The results are demonstrated in the following table, showing that fine tuning the vision encoder performs similar to without finetuning the model, with only 1.1 acc performance drop in average of all the VQA evaluation. The performance drop might due to the challenge of  optimizing the coupled vision encoder and large language model.
>
> | Model                              	| OKVQA |  GQA  | vizwiz |  VSR  | IconVQA |  HM   | average |
> |----------------------------------------|:-----:|:-----:|:------:|:-----:|:-------:|:-----:|:-------:|
> | Finetuning the vision encoder      	| 51.6  | 53.8  |  27.6  | 56.3  |  45.3   | 57.6  |  48.7   |
> | Without finetuning the vision encoder  | 52.1  | 54.6  |  29.4  | 59.9  |  45.6   | 57.4  |  49.8   |

---

> > ### Comment · Reviewer_WwXo · 2023-11-23
> > **thanks!**
> >
> > thanks for these experiments! I think these will improve the paper if included. I still vote to accept the paper :)
> >
> > I read the other reviews too and would still vote to accept this paper. In particular I think the provided ablations answer AcHr's concerns. I'm also okay with the higher level story about a general model outperforming task specific ones and so I'm not persuaded by FCUs's argument. I thus vote to accept this paper still.

---

### Author Response · Authors · 2023-11-22
**General Response**

We thank the reviewers for their insightful feedback. We are particularly encouraged by their recognition of our project's significant contributions (Reviewer WwXo, AcHr, NiES), their belief that it would greatly benefit the community (Reviewer WwXo, AcHr, NiES), and their positive assessments of its novelty (Reviewer NiES), clarity of methodology (Reviewer AcHr, NiES), comprehensiveness of experiments (Reviewer NiES), and strong model performance (Reviewer NiES).



We have also revised our paper. The changes are highlighted in the blue color. To summarize the modifications:

1. The section on related work has been completely rewritten, also replacing the discussion on advanced language models with a focus on multi-task vision-language models.
2. We have included the baseline models, mPLUG-Owl and Otter, in Table 3 for a comprehensive comparison.
3. The paper now features an expanded discussion on the strengths and limitations of our proposed model.
4. The original supplementary material has been moved into the main body of the paper as an Appendix.
5. In the appendix, we have explained the evaluation metrics and various datasets in more details.
6. We add one demonstration example for ChatBot conversation (in Appendix).

---

### Meta-Review · Area_Chair_HFAo · 2023-12-10

**Metareview:**

The work presents MiniGPT-V2, a multimodal Large Language Model (LLM), showcasing impressive performance in visual question answering and visual grounding benchmarks. This paper has been reviewed by four experts in the field, resulting in a range of reviews that highlight both strengths and areas for improvement. The reviewers have acknowledged the authors for the detailed description of their instruction tuning setup, datasets, and training processes. This level of detail is valuable, facilitating future reproduction and comparison of results.

However, despite these positive aspects, the reviewers have identified critical areas that require further attention. Firstly, there is concern regarding the experimental settings, including aspects like the train/test split. Refining these settings could enhance the robustness and generalizability of the study's results. Secondly, the comparisons made with recent baselines have not fully convinced the reviewers of the model's superiority or distinctiveness. More rigorous and comprehensive comparisons are essential to clearly establish the model's advancements over existing methods.

While the paper undoubtedly contributes to the field, the decision, considering the reviewers' feedback, leans towards not recommending acceptance in its present form. We encourage the authors to diligently address these concerns, particularly focusing on improving the experimental settings and strengthening the comparative analysis. The potential of MiniGPT-V2 is evident, and with careful revision, particularly in the areas highlighted by the reviewers, the paper could be well-suited for future submission.

**Justification For Why Not Higher Score:**

1) The experimental settings (e.g., train/test split) could be improved. 2) The comparisons with recent baselines are not convincing enough.

**Justification For Why Not Lower Score:**

N/A

---

### Decision · Program_Chairs · 2024-01-16

Reject